# Domain-interface dynamics of CFTR revealed by stabilizing nanobodies

Maud Sigoillot[1], Marie Overtus[1], Magdalena Grodecka[1], Daniel Scholl[1], Abel Garcia-Pino [2], Toon Laeremans[3,4], Lihua He[5], Els Pardon [3,4], Ellen Hildebrandt[6], Ina Urbatsch[6], Jan Steyaert [3,4], John R. Riordan[5] & Cedric Govaerts[1]

The leading cause of cystic fibrosis (CF) is the deletion of phenylalanine 508 (F508del) in the first nucleotide-binding domain (NBD1) of the cystic fibrosis transmembrane conductance regulator (CFTR). The mutation affects the thermodynamic stability of the domain and the integrity of the interface between NBD1 and the transmembrane domain leading to its clearance by the quality control system. Here, we develop nanobodies targeting NBD1 of human CFTR and demonstrate their ability to stabilize both isolated NBD1 and full-length protein. Crystal structures of NBD1-nanobody complexes provide an atomic description of the epitopes and reveal the molecular basis for stabilization. Furthermore, our data uncover a conformation of CFTR, involving detachment of NBD1 from the transmembrane domain, which contrast with the compact assembly observed in cryo-EM structures. This unexpected interface rearrangement is likely to have major relevance for CF pathogenesis but also for the normal function of CFTR and other ABC proteins.

---

[1] SFMB, Université Libre de Bruxelles (ULB), CP206/02, Boulevard du Triomphe, building BC, B-1050 Brussels, Belgium. [2] Laboratoire de Microbiologie Moléculaire et Cellulaire, ULB CP300, rue des Professeurs Jeener et Brachet 12, B-6041 Charleroi, Belgium. [3] Structural Biology Brussels, Vrije Universiteit Brussel (VUB), Pleinlaan 2, B-1050 Brussels, Belgium. [4] VIB-VUB center for Structural Biology, VIB, Pleinlaan 2, B-1050 Brussels, Belgium. [5] Department of Biochemistry and Biophysics and Cystic Fibrosis Center, University of North Carolina-Chapel Hill, Chapel Hill, NC 27599, USA. [6] Department of Cell Biology and Biochemistry and Center for Membrane Protein Research, Texas Tech University Health Sciences Center, 3601 4th Street, Stop 6540, Lubbock, TX 79430, USA. Correspondence and requests for materials should be addressed to C.G. (email: Cedric.Govaerts@ulb.ac.be)

Cystic fibrosis (CF), is a life-threatening autosomal recessive genetic disorder that originates from mutations in the cystic fibrosis transmembrane conductance regulator (CFTR) gene[1]. The most prevalent mutation is the deletion of a phenylalanine residue at position 508 (F508del) within the first nucleotide-binding domain (NBD1). This mutation is responsible for misfolding and early degradation of CFTR, leading to disruption of ionic and water homeostasis in epithelial cells of various organs such as lungs, pancreas, and intestine.

Although it functions as an ion channel, CFTR belongs to the ATP-binding cassette (ABC) transporter superfamily from a structural and evolutionary standpoint. The recent electron cryo-microscopy (cryo-EM) structures of zebrafish and human CFTR[2–4] have confirmed that it adopts the common architecture of ABC proteins, with 12 transmembrane helices (TMDs) and two nucleotide-binding domains (NBDs) located in the cytoplasm. The NBDs are connected to the TMDs by short coupling helices named intracellular loops (ICLs) and F508 in NBD1 makes contact with ICL4. Interestingly, while crystallographic studies indicated that isolated NBDs can form well-structured domains[5–7], they appear less defined than the TMDs in the cryo-EM structures, with higher B factors which may reflect the dynamic character of these regions. Another hallmark of CFTR is its additional dynamic cytoplasmic domain, named R domain[8] that controls channel gating in response to phosphorylation by protein kinase A (PKA). The R domain is only partly seen on the cryo-EM structure of dephosphorylated CFTR[2] and appears to be located between the TMDs. CFTR also contains a 32-residue segment termed the regulatory insertion (RI), located in position 405–436 in NBD1, not present in other ATP-binding cassette transporters. Removal of RI enables F508del CFTR to mature and traffic to the cell surface where it mediates regulated anion efflux and exhibits robust single chloride channel activity[9].

Over 300 cystic fibrosis-causing mutations have been described in the CFTR gene (The Clinical and Functional TRanslation of CFTR (CFTR2); available at http://cftr2.org), and they are spread over various parts of the protein, indicating that several pathogenic mechanisms are possible. It is now recognized that the intrinsic dynamics and relative instability of the protein are central elements in the physiopathology of CF. The most dramatic illustration of this behavior is that the deletion of F508 perturbs NBD1 thermodynamic stability, and the interface between the NBDs and the TMDs[10–12]. This deleterious effect can be compensated by a variety of mutations in NBD1 at different locations. Introducing such stabilizing mutations in a F508del CFTR background permits maturation of a functional channel[5,6,13]. Remarkably, the extent of recovery in protein maturation seems to be directly proportional to the ability of specific compensating mutations to increase thermal stability of NBD1[14]. To date, current correctors of CFTR have a limited ability to improve patient's conditions and their mechanisms of action remain poorly understood. This is particularly true for correctors of F508del mutation promoting CFTR maturation without restoration of its thermodynamic stability[15]. In this context, developing NBD1-specific chaperones with the ability to improve the thermostability of F508del mutant would provide a promising route for effective therapy.

To study CFTR conformational dynamics and engage an approach for CFTR stabilization from a therapeutic point of view, we turned to nanobodies, which are the antigen-binding fragments of heavy-chain-only antibodies found in camelids[16]. This approach stemmed from the recognized ability of nanobodies to thermally stabilize a specific conformation of their target antigen, including in the case of highly dynamic membrane proteins[17,18]. We report here the development of nanobodies against NBD1 of human CFTR, characterize their interaction with the target, and demonstrate the ability of several of them to thermally stabilize CFTR by 10 °C or more. High-resolution structures of the complexes reveal the mechanism of stabilization at the molecular level, providing possible routes for development of therapeutic stabilizers. Furthermore, the location of several epitopes demonstrates that CFTR must be able to adopt at least one additional conformation that differ significantly from the published cryo-EM structures, consistent with the view that CFTR is a highly dynamic protein, even under a normal physiological regime.

## Results

**Generation of nanobodies against NBD1.** Two different llamas were immunized with 2PT-NBD1, a stabilized version of human CFTR NBD1 domain bearing the mutations S492P/A534P/I539T[19]. Nanobodies were obtained after phage display selection, using established protocols[16]. After two rounds of selection against 2PT-NBD1, a set of candidate binders was isolated. Among these nanobodies, we focused our biochemical characterization effort on 5 different nanobodies belonging to different sequence clusters, classified according to the sequences of the third complementarity determining region (CDR3) (Supplementary Fig. 1).

**Nanobodies bind NBD1 with high affinity.** Specific binding and apparent affinity of purified nanobodies D12, T2a, T27, T4, T8, and G3a to 2PT-NBD1 were confirmed by enzyme-linked immunosorbent assay (ELISA) using immobilized 2PT-NBD1. Dose–response curves indicated a highly potent binding to 2PT-NBD1 and F508del-2PT-NBD1 ($EC_{50}$ in the 1–50 nM range) for all nanobodies, except nanobody G3a, which displayed a weaker binding potency (Fig. 1a, e). Interestingly, F508del mutation drastically affected binding of nanobodies T4, T8 and to a lesser extent G3a, while nanobodies D12, T2a and T27 were not affected by the deletion (Fig. 1b–d).

Isothermal titration calorimetry (ITC) was used to characterize the thermodynamic parameters of the binding (Fig. 1e and Supplementary Fig. 2). In each case the titrations were consistent with a 1:1 bimolecular association between nanobodies and monomeric 2PT-NBD1 with $K_D$ values of $54 \pm 10$, $25 \pm 10$, $37.9 \pm 7$, and $39 \pm 2$ nM with nanobodies T2a, T27, T4, and T8, respectively. Nanobody G3a bound 2PT-NBD1 with a lower affinity ($1.1 \mu M \pm 0.1 \mu M$), which is consistent with our determination of apparent affinity by ELISA (Fig. 1a, e). These $K_D$ values are similar to those measured for other in vivo matured camelid heavy-chain antibodies interacting with their ligands[20]. The thermodynamic parameters of the interaction of nanobodies with 2PT-NBD1 were determined based on these ITC measurements (Fig. 1e and Supplementary Fig. 2).

**Thermal stabilization of NBD1 by nanobodies.** A range of mutations in NBD1 has already been described to improve thermal stability of this domain[5,9,21] and more recently, He et al., showed that small molecules such as indole based-compounds, can counteract the folding defect of CFTR by stabilizing NBD1[14]. In the context of F508del, specific NBD1 chaperones could be key molecules to overcome the deleterious effects of the mutation from a therapeutic perspective[22]. We therefore evaluated the effect of our different nanobodies on apparent thermal unfolding (Tm) of 2PT-NBD1 by thermal shift assay (illustrated in Fig. 2a and Supplementary Fig. 3). As shown in Fig. 2b, binding of nanobodies D12, T2a, T27, T4 and T8 led to strong stabilization of 2PT-NBD1, with increases of Tm of 13.8, 11.2, 11.9, 12.5 and 9.1 °C, respectively. In contrast, G3a did not induce a significant shift in 2PT-NBD1 thermal stability. We confirmed the stabilizing

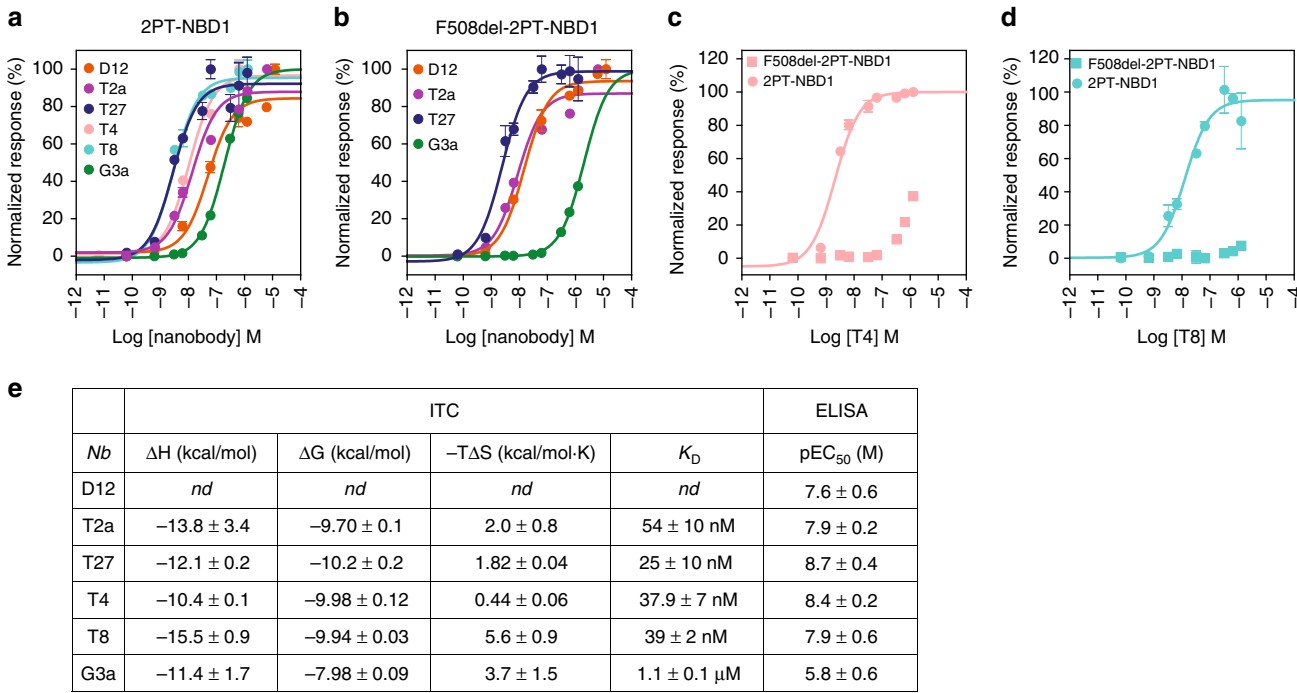

**Fig. 1** Binding of NBD1-specific nanobodies to isolated 2PT-NBD1 and F508del-2PT-NBD1. **a** Nanobody binding to 2PT-NBD1 measured by ELISA. Biotinylated 2PT-NBD1 was immobilized on neutravidin-coated plates and incubated with increasing concentration of each nanobody. Binding of nanobody was followed by immunodetection of the His$_6$-tag (see Methods). Representative curve of 3 independent experiments is shown. Error bars represent the standard deviation (SD) of duplicates. Data were normalized to maximum signal for each nanobody separately. **b** Nanobody binding to F508del-2PT-NBD1 measured by ELISA as described in (**a**). **c** Nanobody T4 binding to 2PT-NBD1 and F508del-2PT-NBD1. Data were normalized to maximum signal of T4 binding to 2PT-NBD1. **d** Nanobody T8 binding to 2PT-NBD1 and F508del-2PT-NBD1. Data were normalized to maximum signal of T8 binding to 2PT-NBD1. **e** Thermodynamic parameters of nanobody binding to 2PT-NBD1 determined using isothermal calorimetry (curves shown in Supplementary Fig. 2), and pEC$_{50}$ determined by ELISA (**a**). $K_D$ values determined by ITC represent mean ± SD ($n = 3$). Source data are provided as a Source Data file

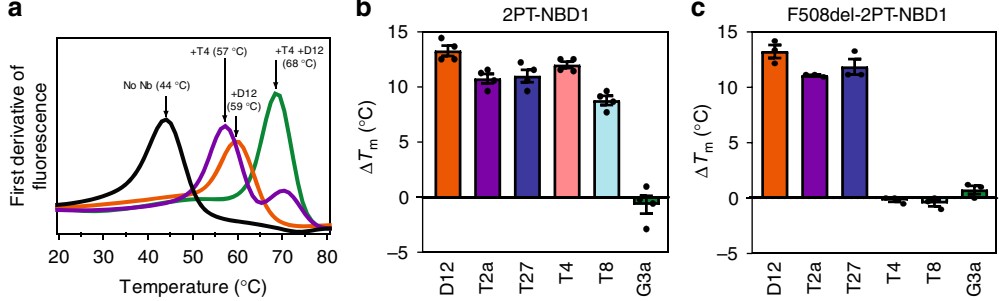

**Fig. 2** Stabilization of isolated 2PT-NBD1 and F508del-2PT-NBD1 variants by nanobodies. **a** Differential scanning fluorescence (DSF) of purified 2PT-NBD1. The protein (alone or in complex with one or two different nanobodies) was incubated with SYPRO Orange dye and fluorescence was measured as a function of temperature. The melting temperatures (Tm) were determined by the maxima of the first derivative of fluorescence. Curves depict mean of duplicates of one experiment representative of at least three independent experiments. **b** Summary of melting temperature differences (ΔTm) of 2PT-NBD1 in presence of different nanobodies determined using DSF as in (**a**). Data are mean ± SEM of duplicates from four independent experiments. **c** Summary of melting temperature differences (ΔTm) of F508del-2PT-NBD1 in presence of different nanobodies using DSF as in (**a**). Data are mean of duplicates ± SEM ($n = 4$). Source data are provided as a Source Data file

effect using differential scanning calorimetry (DSC) for two nanobodies (T2a and T8) and obtained similar increases in Tm (Supplementary Fig. 3). The stabilizing effects of D12 and T4 nanobodies were additive in combination, leading to an apparent melting temperature of the complex of 68 °C, which is 24 °C higher than isolated 2PT-NBD1 (Fig. 2a). Interestingly, nanobodies D12, T2a and T27 stabilized F508del-2PT-NBD1 mutant to the same extent as 2PT-NBD1. In agreement with our binding data (Fig. 1c, d), nanobodies T4 and T8 did not stabilize F508del-2PT-NBD1 (Fig. 2c).

**Structures of nanobody-NBD1 complexes.** In order to identify the molecular basis of NBD1 stabilization by nanobodies, we determined the crystal structure of each complex. Crystallization trials were performed using either 2PT-NBD1 or ΔRI-NBD1 constructs, as removal of the RI is known to improve the protein stability and favor crystallogenesis[7]. We employed various crystallization strategies, using multiple stabilizing nanobodies at the same time and/or limited proteolysis to facilitate crystal formation. Structures were solved by molecular replacement using published structures of human NBD1 and nanobodies.

### Table 1 Data collection and refinement statistics (molecular replacement)

|  | ΔRI-NBD1-D12-T4 PDB: 6GJS | 2PT-NBD1-T2a-T4 PDB: 6GJU | 2PT-NBD1-T27 PDB: 6GJQ | ΔRI-NBD1-D12-T8 PDB: 6GK4 | ΔRI-NBD1-D12-G3a PDB: 6GKD |
|---|---|---|---|---|---|
| *Data collection* |  |  |  |  |  |
| Space group | C 1 2 1 | C 2 2 2₁ | P 2₁ 2₁ 2₁ | P 1 2₁ 1 | P 2₁ 2₁ 2₁ |
| Cell dimensions |  |  |  |  |  |
| $a, b, c$ (Å) | 152.2, 41.6, 99.3 | 38.68, 135.78, 190.65 | 64.49, 118.15, 180.21 | 80.94, 55.19, 114.99 | 116.94, 146.83, 188.34 |
| $\alpha, \beta, \gamma$ (°) | 90, 120.56, 90 | 90, 90, 90 | 90, 90, 90 | 90, 103.96, 90 | 90, 90, 90 |
| Resolution (Å) | 42.6–1.95 (2.02–1.95) | 46.4–2.6 (2.69–2.6) | 45.05–2.49 (2.58–2.49) | 45.16–2.91 (3.01–2.91) | 34.43–2.99 (3.10–2.99) |
| $R_{sym}$ or $R_{merge}$ | 0.07603 (0.584) | 0.1204 (1.649) | 0.1819 (1.64) | 0.135 (0.592) | 0.155 (1.106) |
| $I/\sigma I$ | 11.82 (2.32) | 12.13 (0.99) | 10.34 (1.23) | 8.1 (2.2) | 9.01 (1.25) |
| Completeness (%) | 99 (98) | 100 (100) | 99 (93) | 99 | 99 (93) |
| Redundancy | 3.7 (3.7) | 6.4 (6.5) | 9.1 (8.7) | 3.3 (3.5) | 4.5 (4.1) |
| *Refinement* |  |  |  |  |  |
| Resolution (Å) | 42.6–1.95 (2.02–1.95) | 46.4–2.6 (2.69–2.6) | 45.05–2.49 (2.58–2.49) | 45.16–2.91 (3.01–2.91) | 34.43–2.99 (3.10–2.99) |
| No. reflections | 144455 | 103077 | 445126 | 72866 | 293796 |
| $R_{work}/R_{free}$ | 0.179/0.213 | 0.223/0.247 | 0.197/0.242 | 0.246/0.295 | 0.204/0.235 |
| No. atoms | 3931 | 3254 | 10055 | 6870 | 20899 |
| Protein | 458 | 421 | 1231 | 894 | 2761 |
| Ligand/ion | 33 | 13 | 124 | 70 | 323 |
| *B*-factors (Å²) |  |  |  |  |  |
| Protein | 34 | 78 | 51 | 52 | 76 |
| Ligand/ion | 39 | 81 | 84 | 77 | 95 |
| R.m.s. deviations |  |  |  |  |  |
| Bond lengths (Å) | 0.014 | 0.012 | 0.015 | 0.015 | 0.013 |
| Bond angles (°) | 1.68 | 1.66 | 1.97 | 1.82 | 1.66 |

Values in parentheses are for highest-resolution shell

Resolutions ranged from 1.9 to 3.0 Å (Table 1), allowing complete description of the binding interfaces.

As listed in Table 1, by multiplying nanobody combinations to help crystal formation, we solved the different interfaces several times under various crystallization conditions and in all cases showing a similar binding mode for a given nanobody. For example nanobody D12 was observed in 3 different crystals: the ΔRI-D12-T4 complex (diffracting up to 1.95 Å), the ΔRI-D12-T8 complex (diffracting to 2.90 Å) and the ΔRI-D12-G3a complex treated with papain (diffracting to 3.00 Å). Comparison of the NBD1-D12 interfaces (residues with atoms closer than 4 Å) leads to $C_\alpha$-atom root mean square deviations (RMSDs) below 0.41 Å.

In some cases, in situ limited proteolysis using papain or subtilisin A was required to generate diffracting crystals. Limited proteolysis is typically used to remove flexible loops that can prevent lattice formation[23]. Analysis of the structures showed that the nanobodies themselves remained unaffected by protease treatment and that the complete binding interface is present and clearly seen in all structures. In contrast, significant portions of the NBD1 domain were cleaved, but the remaining fragment exhibited the typical NBD1 fold, albeit with some minor deviations far from the binding interface. For example, in the 2PT-NBD1-T2a-T4, papain cleavage at position K447 of 2PT-NBD1 led to the crystallization of a fragment of 2PT-NBD1 missing residues 389–447 and no electron density was observed for the likely flexible C terminal segment 638–646 and the loop 479–483 from the ABCβ subdomain. Nevertheless, the overall folding of 2PT-NBD1 polypeptide was highly similar to that of the previously published structure of NBD1 (PDB: 2PZE) with a root mean square deviation for the α-carbons of 0.78 Å. In addition, the regions involved in binding are highly similar in spite of protease treatment. For example we measured an overall RMSD below 1 Å for the Cα's of residues involved in the binding interface shared (see below) by D12 (where no protease was used), T2a (treated with papain) and T27 (treated with subtilisin). Analysis of the different crystal structures revealed that 3 different epitopes are recognized by the 5 nanobodies characterized here.

**A first stabilizing epitope includes several subdomains**. Nanobodies D12, T2a, and T27 recognize the same epitope (Fig. 3a) located on the edge of the α/β-core region, including the first residues of the Walker A motif and the last residues of the Walker B (Fig. 3b). Although these nanobodies belong to different

sequence clusters (Supplementary Fig. 1), their mode of binding is remarkably similar. While nanobodies typically recognize their cognate epitope via their highly variable and long CDR3[20,24], these three nanobodies interact with NBD1 not only through residues from CDRs but also through their (conserved) framework regions. This observation explains the particularly large binding interfaces, extending over 1000 Å², with multiple contacts across the interface conserved among the nanobodies.

In each of these three nanobodies, the CDR3 adopts a β-strand configuration, further extending the overall β-sandwich fold of the nanobody. The CDR3's contain one of two acidic residues that form an ionic interaction with K606 (Fig. 3c) in NBD1. Hydrogen bonds are formed between acidic side-chains and backbone amides, for example E608 in NBD1 with backbone from D109 in T27 or from D111 in D12 (illustrated in Fig. 3c), forming a tight set of polar interactions together with the aforementioned ionic bond.

A set of hydrophobic interactions are observed towards the tip of the CDR3 loops of these nanobodies. This loop sits on top of the Walker A motif, where hydrophobic side chains from the nanobody occupy a small cavity present in the neighboring α/β-subdomain (see L108 in D12, Fig. 3c).

We observe interaction between NBD1 and sidechains from the framework of the nanobody such as the conserved Y37 that forms a H-bond with the backbone amide from V580 in all of the structures of these three nanobodies. In addition, a hydrogen bond is observed between an Asp found at the tip of the CDR1 of nanobodies D12 and T2a and the backbone amides of G550 and G551 (slight differences are seen between the different solved structures). In summary, for these three nanobodies, the large interface can be similarly decomposed into four main contact sites, where specific interactions (electrostatic, hydrophobic and H-bond) are formed, extending over different subdomains of NBD1, covering over 30 Å in its longest axis.

The location of these nanobodies completely overlaps (Supplementary Fig. 4a) with that of the C-terminal regulatory extension (RE) observed in previously published structures of NBD1[25,26]. The RE segment, comprising residues 654–673 was removed from the 2PT-NBD1 construct used for immunization and characterization. While the functional role of the RE is still unclear, it has been described to be a very mobile domain[27]. When we tested whether our nanobodies were able to bind a construct containing the RE (2PT-NBD1-RE), we still observed high-affinity binding for nanobodies D12, T2a and T27, albeit with decrease in

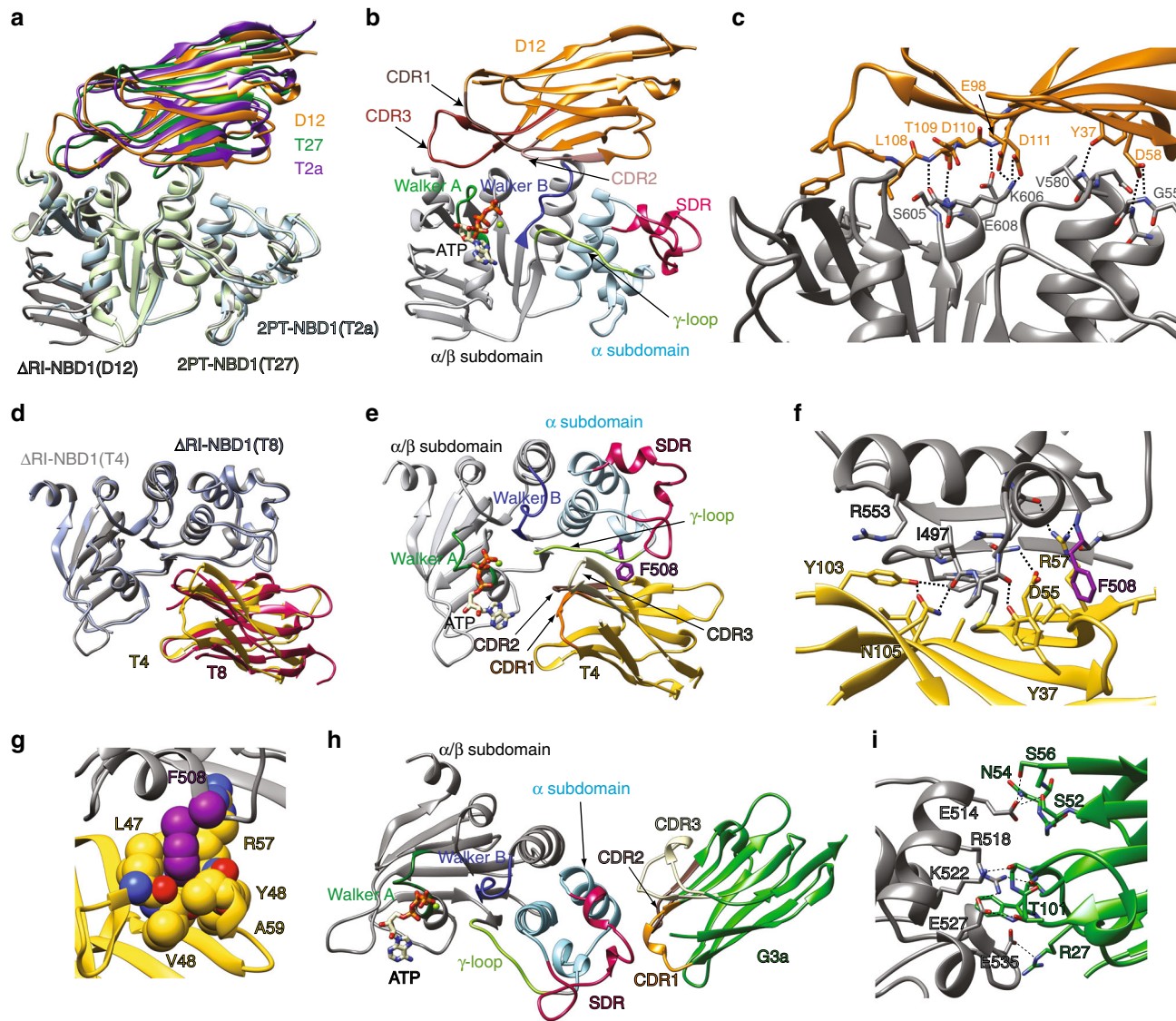

**Fig. 3** Crystal structures of NBD1-nanobody complexes. **a** Structures of nanobodies D12, T2a and T27 bound to NBD1. Superimposition was performed on the NBD1 region. **b** Structure of NBD1-nanobody D12 complex highlighting the different structural elements of hNBD1 as well as the CDRs of the nanobody. **c** Details of the interface between nanobody D12 and NBD1. Polar interactions are highlighted by dashed lines (see text for description). Only side chains participating in the interface are explicitly shown. **d** Structures of nanobodies T4 and T8 bound to NBD1. Superimposition was performed on the NBD1 region. The view is rotated compared to (**a**). **e** Structure of NBD1-T4 nanobody complex highlighting the different structural elements of NBD1 as well as the CDRs of the nanobody **f** Details of the interface between nanobody T4 and NBD1. Polar interactions are highlighted by dashed lines (see text for description). Only side chains participating in the interface are explicitly shown. **g** Close-up of the interaction of F508 from NBD1 to residues in T4. Atoms are shown as space-filling model to highlight the contacts, occurring at van der Waals distances. **h** Structure of NBD1-G3a nanobody complex **i** Details of the interface between nanobody G3a and NBD1. Polar interactions are highlighted by dashed lines (see text for description). Only side chains participating in the interface are explicitly shown

apparent $EC_{50}$ compared to 2PT-NBD1 (Supplementary Fig. 4b). This is consistent with RE being a dynamic region of CFTR.

**A second stabilizing epitope includes F508.** Although nanobodies T4 and T8 share no sequence similarity in CDR3, the crystal structures revealed that they bind NBD1 in the same location, a groove which includes the γ-phosphate switch loop/Q-loop (Fig. 3d, e) with an overall binding interface of over 900 Å². Here also we observed a non-classical nanobody-antigen-binding mode in which the CDR3s contributed only a portion of the interface. Close inspection revealed that for both nanobodies 3 hydrogen bonds are formed between CDR3 residues and the Q-loop backbone, for example between Y103 and N105 in T4 and

the carbonyl of I497 (Fig. 3f). Y103 is also interacting with R553 in the α-subdomain of NBD1 through cation-π interactions.

The other CDRs also participate in the interface, including a salt bridge observed between D54 in the CDR2 of T8 and K564 of NBD1, and a hydrogen bond observed between D55 of T4 and the backbone amide of F490. The conserved Y37 is also participating in the interface, in this case with the backbone carbonyl of P499. R57 in T4 (R58 in T8) makes a hydrogen bond with the backbone carbonyl of R560 and importantly also with the backbone carbonyl of F508.

Indeed, one of the key features of the T4/T8 interface is that it directly involves F508. As shown in Fig. 3g, F508 is nestled inside a hydrophobic pocket formed by residues located between the second framework β-strand and CDR2, namely P47, L50, A60

and the $C_\beta$ of R58 in the case of T8, while for T4 the pocket is made up of L47, V50, A59, backbone atoms of V48 and Y48 as well as the $C_\beta$ of R57. In both cases the carbons of the aromatic ring of F508 are in ideal proximity to these side chains to form van der Waals interactions. Therefore, F508 is clearly part of the binding interface and it is thus not surprising that the binding of both T4 and T8 to F508del-2PT-NBD1 is drastically affected by F508 deletion (Fig. 1c, d).

**Nanobody G3a recognizes the structurally diverse region**. The non-stabilizing nanobody G3a (Fig. 2b) recognizes a third epitope located entirely in the so-called structurally diverse region (SDR) of NBD1 (Fig. 3h) with an overall surface of about 650 Å$^2$. On the nanobody, residues from the three CDRs (but not from the framework regions) contribute a series of hydrogen bonds (Fig. 3i). Residues S52, N54 and S56 in CDR2 form a tight cluster of hydrogen bonds with E514. CDR3 residues are involved in only two contacts (hydrogen bonds with K522 and E527), while CDR1 interacts more extensively, in particular as the formation of a short α-helix allows W31 to form cation-π interaction R518, which itself interacts with the backbone carbonyl of W31, and a salt bridge is observed between E535 from NBD1 and R27 from CDR1.

This third epitope solely involves a single subdomain (spanning between residues 514 and 535), located on the tip of NBD1, unlike the other two epitopes in which the stabilizing nanobodies contact residues located far apart in NBD1, thus likely reducing conformational flexibility of NBD1.

**Interaction of nanobodies with full-length CFTR**. As discussed above, thermal stabilization of NBD1 may provide a novel therapeutic route against the destabilizing F508del mutation. Considering that the stabilizing nanobodies described here were developed using isolated recombinant NBD1 domain for both immunization and selection, we investigated the ability of the nanobodies to recognize and stabilize the full-length CFTR (FL-CFTR).

We thus tested the ability of these nanobodies to bind FL-CFTR in different assays. First, we used purified human CFTR to quantify binding potencies of representative nanobodies (one for each epitope) in an ELISA assay. When the nanobodies were immobilized and purified FL-CFTR was titrated, T2a, T8 and G3a were all able to bind with high affinity (Fig. 4a) reaching similar $B_{max}$ values, demonstrating that each of the three epitopes identified was accessible in the context of the full-length protein. Interestingly, when performing the assay using immobilized CFTR and titrating the nanobodies (Fig. 4b), we observed that G3a and T8 reached $B_{max}$ values lower than that observed for T2a (about 50 and 30% of T2a maximum signal respectively, Fig. 4c). This indicates that the epitopes of these two nanobodies are not accessible in a subset of population, suggestive of conformational diversity in the ensemble. We then used flow cytometry on permeabilized baby hamster kidney (BHK)-21 cells stably expressing human CFTR to establish whether the different nanobodies were capable of recognizing FL-CFTR in a cellular context. When comparing the fluorescence measured for the NBD1-specific nanobodies to that of the negative control (irrelevant nanobody, i.e. directed against a non-CFTR antigen) we observed strong increase in median signal, ranging from 3 fold to 5 fold over control (Fig. 4e, f), demonstrating all of these nanobodies also bind cellular CFTR. A similar behavior was observed for D12, T27 and T4 nanobodies (Supplementary Fig. 5b, c). In order to test whether this signal was originating from binding to mature FL-CFTR, we performed pull-down of cellular CFTR with T2a, T8 and G3a nanobody and analyzed the

isolated CFTR by immunodetection after electrophoresis. Functional mature CFTR being fully glycosylated, electrophoresis allows to separate it from the intracellular immature CFTR. Mature CFTR with complex N-linked oligosaccharide chains migrates at an apparent molecular weight of 170 kDa (historically called band C) while immature core-glycosylated CFTR runs at a lower molecular weight (named band B). As shown in Fig. 4g, immunoblot analysis indicated that CFTR recognized by the three nanobodies shows an identical electrophoresis pattern as observed in whole cell lysate, where the large majority of the protein migrates to an apparent size of 170 kDa, which is expected for glycosylated CFTR (band C, highlighted in Fig. 4g), and thus mature protein. This was also observed for D12, T27 and T4 nanobodies (Supplementary Fig. 5d).

In order to verify that the recognition of FL-CFTR by the nanobodies followed the binding modes observed on isolated NBD1, we performed flow cytometry experiment on 2PT-F508del expressing cells. This version of F508del is stabilized by three point mutations (I539T/S492P/A534P) which enable proper folding and maturation of CFTR, leading recovery of channel activity[19]. As shown in Fig. 4h, i, nanobodies T2a and G3a bind efficiently this mutant, indicating that the native fold of NBD1 is present. However, nanobody T8 is not able to bind this mutant, most likely due to the lack of F508, which is involved in its epitope (Fig. 3g) and thus also required for binding of T8 to isolated NBD1 (Fig. 1d).

As our nanobodies are directed against NBD1 and that current models suggest that NBD1 and NBD2 must make contact in order to hydrolyze ATP[4,8], we tested whether the different nanobodies could affect ATPase activity of CFTR. Incubation with saturating concentration of nanobodies D12, T2a, and T27 and strongly reduced ATPase activity to respectively 50, 50 and 30% of PKA-phosphorylated CFTR (Fig. 5a), demonstrating nanobody interaction with the active, phosphorylated protein. ATPase activity was lowered to 60% in presence of nanobody T8, while G3a did not affect it.

ATPase activity was also used to measure thermal inactivation of CFTR, an assay shown to coincide with thermostability of NBD1[28]. Addition of each of the different nanobodies shifted CFTR inactivation to higher temperature, up to 7 °C for the best stabilizing nanobody D12 (Fig. 5b), just as these nanobodies increased the apparent Tm of isolated NBD1 (Fig. 2b). We noted that, while ATPase activity of CFTR was not affected by the presence of G3a, thermal inactivation was shifted by 3.1 °C in the presence of G3a (Fig. 5b). This contrasts with the behavior observed by thermal shift assay where G3a did not affect the apparent Tm of isolated NBD1 (Fig. 2b, c).

Nanobody stabilization of human FL-CFTR was confirmed with nanoscale differential scanning fluorimetry (nanoDSF). The analysis was performed with a stabilized version of human CFTR (stab-CFTR: 2PT/ΔRI/R1048A_1172X), allowing the production and purification of sufficient amount of functional human CFTR in detergent. As illustrated by the melting curves of stab-CFTR alone or in complex with T2a or T4 (Fig. 5c, d), we observed thermostabilization of 8 °C, which is an example of a CFTR-specific reagent with strong stabilizing properties. Tm values obtained by nanoDSF are summarized in Fig. 5e.

## Discussion
While remarkable progress has been made in the development of CFTR correctors in the last few years, little or no mechanistic insights are available to rationalize their mode of action. On the other hand, it is known that the destabilizing effect of F508del can be compensated by artificially introduced specific mutations at various sites in NBD1, leading to significant recovery of channel

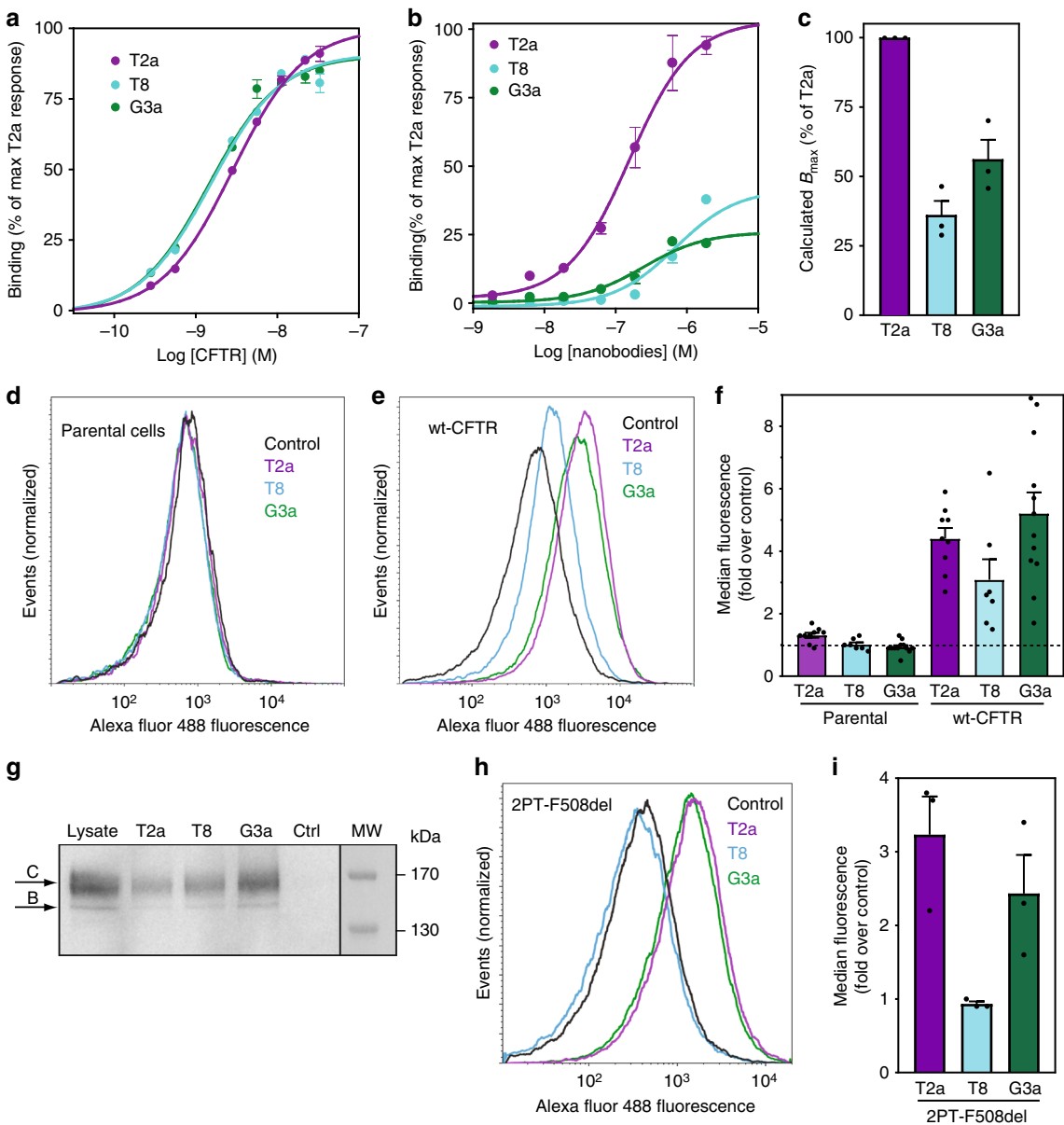

**Fig. 4** Binding of nanobodies to FL-CFTR. **a** Dose–response ELISA of interactions between wt-CFTR and nanobodies. Immobilized nanobodies (T2a, T8 and G3a) were incubated with different concentrations of purified CFTR. **b** Immobilized purified wt-CFTR was incubated with increasing concentrations of nanobodies. For both (**a**, **b**) Data were normalized to maximal response of T2a after subtraction of the signal from the negative control nanobody. Graph depicts one representative of at least three independent experiments. Error bars represent standard deviations of triplicates. **c** Average $B_{max}$ of 3 independent experiments (±SEM) calculated for curves in (**b**). **d** Flow cytometry analysis of nanobodies T2a, T8 and G3a on parental BHK-21 cells shows no difference in labelling compared to a negative control nanobody while in (**e**) increased labelling is observed for the NBD1-specific nanobodies in BHK-21 cells overexpressing wt-CFTR. Data were normalized to the number of events acquired in each condition. Graph depicts one representative of at least three independent experiments. **f** Average median fluorescence (fold over negative control) for each of the three representative nanobodies as illustrated in (**d**, **e**). Quantification of at least 3 independent experiments (±SEM). **g** Immunoblot of CFTR from solubilized BHK-21 cells pulled-down with His$_6$-tagged nanobodies, including a non-CFTR nanobody as a control. Eluted nanobodies-CFTR complexes were separated by SDS-PAGE and presence of CFTR was detected with mAb 596 antibody after immunoblotting. Arrows indicate the mature (band C) and immature (band B) forms of CFTR. Representative of at least 3 independent experiments. **h** Flow cytometry analysis of nanobodies T2a, T8 and G3a on BHK-21 cells expressing 2PT-F508del showing increased labelling for T2a and G3a, but not T8. **i** Quantification of data illustrated in (**h**). Average of 3 independent experiments (±SEM). Source data are provided as a Source Data file

expression and activity[19,29]. This shows that the molecular stress caused by F508del can be counteracted allosterically and the development of NBD1 chaperones remains an underexplored therapeutic route. Indeed, so far, few molecules have been shown to specifically stabilize CFTR or even NBD1[14]. Studies have shown that small compounds such as BIA or BEIA are able to slightly stabilize the protein (<3 °C) but only at very high

concentrations (close to mM)[14], thus precluding any therapeutic developments[26].

This study demonstrates that large thermal stabilization (>10 °C) of isolated NBD1 and of full-length CFTR can be achieved with antibodies. The stabilizing nanobodies bind distinct, conformational and non-overlapping epitopes, with common features. For instance, the interaction interfaces span several

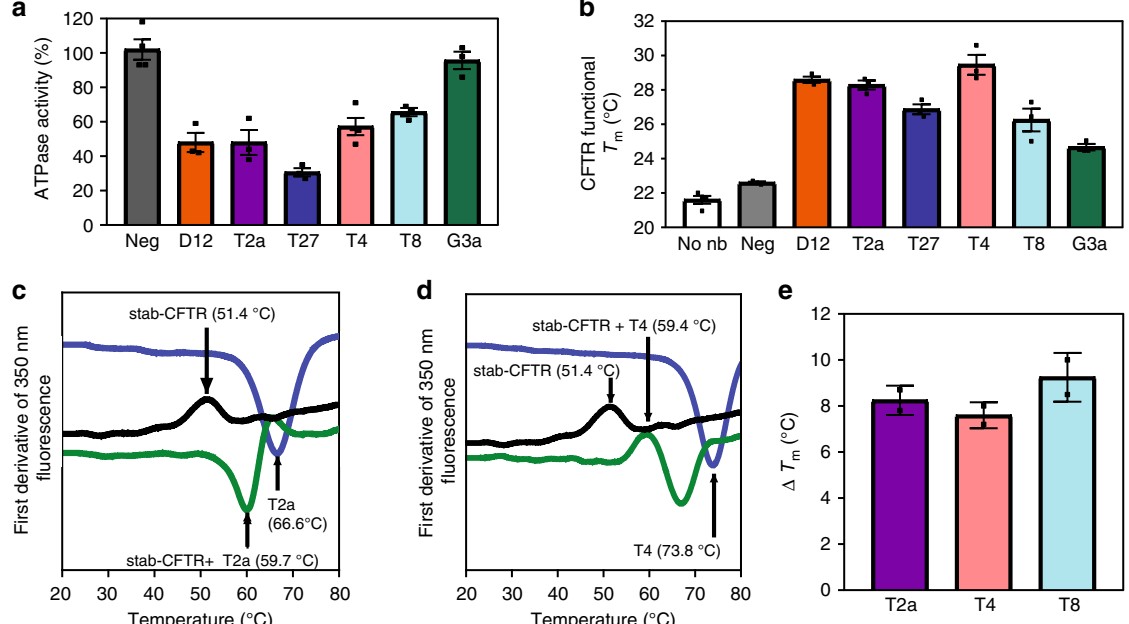

**Fig. 5** Nanobodies reduce ATPase activity of CFTR but increase the temperature of thermal inactivation. **a** Influence of nanobody addition on ATPase activity of wt-CFTR. Conversion of α-$^{32}$P-ATP to ADP was measured after 1 h incubation of wt-CFTR with the different nanobodies. Data from replicate determinations are represented as mean ± SEM ($n = 3$, except for ATPase activity of wt-CFTR activity with nanobodies Neg and T4 for which $n = 4$). **b** Thermoprotection of wt-CFTR activity by nanobodies. Inactivation threshold temperatures were determined by measuring residual ATPase activity after 30 min heat challenge at various temperatures. Data from replicate determinations are represented as mean ± SEM ($n = 3$, except for wt-CFTR activity in absence of nanobody for which $n = 4$). **c** Thermostability of stab-CFTR measured by nanoDSF. First derivative of 350 nm fluorescence as a function of temperature showing the determination of Tm of purified stab-CFTR alone (black) or in complex with nanobody T2a (dark grey). Melting curve of nanobody T2a alone is depicted in light grey. One representative experiment shown. **d** Thermostability of stab-CFTR as in (c), in complex with nanobody T4 (dark grey). Melting curve of nanobody T4 alone is depicted in light grey. One representative experiment shown. **e** Summary of melting temperatures of stab-CFTR in complex with nanobodies T2a, T4 and T8 determined by nanoDSF. Data from triplicates are represented as mean ± SD ($n = 2$). Source data are provided as a Source Data file

subdomains of NBD1, covering relatively large distances (over 30 Å). As such, both families of nanobodies provide, upon binding, a physical connection between the α-subdomain and the α/β-subdomain of NBD1 (Fig. 3b, e). This exogenous bridging of NBD1 tertiary structure is likely to be responsible for the large stabilizing effect observed. In contrast, the non-stabilizing nanobody G3a does not mediate long range connection, instead binding solely to a unique subdomain (SDR). Importantly, binding to two different stabilizing epitopes act on the protein in distinct ways, as incubating NBD1 with D12 and T4 produced additive effects (Fig. 2a), suggesting that several sites could be targeted to maximize therapeutic benefit.

While nanobodies targeting extracellular epitopes of proteins involved in human diseases are currently being developed as potential drugs for a variety of human diseases[30], correcting CFTR folding defect requires intracellular action, and most likely at the level of the endoplasmic reticulum and/or the Golgi apparatus. Various tools for intracellular delivery have been developed over the years to introduce proteins (including antibodies) into cells in a functional state[31–33]. Still, the practical applications of intracellular delivery techniques into therapeutics will likely remain a significant challenge in the foreseeable future. As such, the use of small molecule compounds remains the method of choice for intracellular therapeutic targets, as membrane penetration can be an inherent property of the drug-like molecules. While classical experimental and computational approaches have failed to isolate small molecules with sufficient potential to stabilize CFTR[26], the crystal structures of complexes between NBD1 and various stabilizing nanobodies described here may offer a new route for

rational design of CFTR stabilizers. Indeed, the atomic details of the interactions can be used as a molecular template to design molecules that will recapitulate the key features of the NBD1-nanobody interfaces. In the light of recent developments in computational docking and in pharmacophore building, the isolation of small compounds that can mimic protein-protein interface is becoming a realistic strategy. One of the challenges here will be to develop small molecules that will not only bind given subdomains of NBD1, but also form the physical connection across these subdomains, which will probably require chemically linking molecules targeting different subdomains into a chimeric compound.

Our crystal structures of the complexes allow direct modeling of the binding mode of each nanobody to FL-CFTR by superimposing the coordinates of NBD1 on the recently available cryo-EM structures of CFTR[2–4].

These superimpositions show that the epitope recognized by G3a should be accessible in CFTR (Fig. 6a and Supplementary Fig. 6a) with no visible steric hindrance, correlating with the efficient recognition observed in flow cytometry, ELISA and pulldown experiments with this nanobody. Nanobodies D12, T2a, and T27 are predicted to bind CFTR between NBD1 and NBD2 (Fig. 6b). The structure of dephosphorylated human CFTR (PDB: 5UAK) displays sufficient spread between the two NBDs to allow positioning of the nanobody (a slight increase in the opening could be required to alleviate any minor steric overlap). In contrast, the closing of the NBDs observed in the structure of phosphorylated zebrafish CFTR (PDB: 5W81) is expected to prevent binding of such nanobody (Supplementary Fig. 6b). This agrees well with the strong decrease of ATPase activity observed

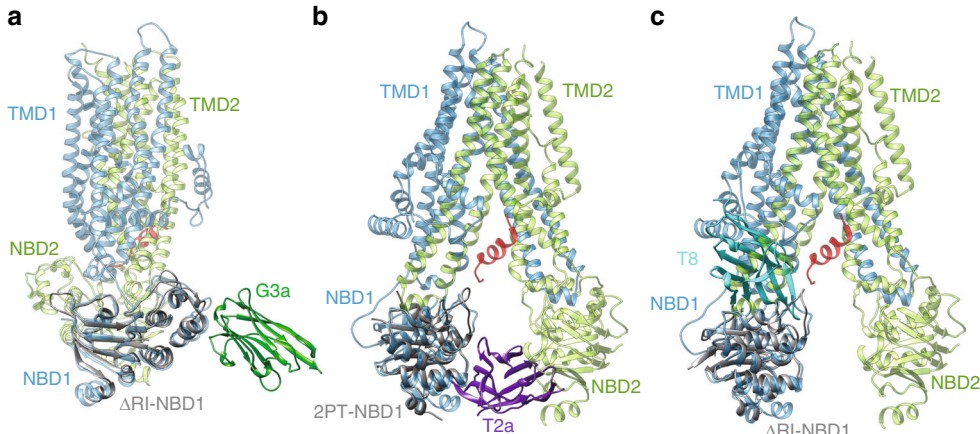

**Fig. 6** NBD1-nanobody complexes superimposed onto the structure of CFTR. **a** The previously reported cryo-EM structure of CFTR (PDB: 5UAK) was aligned with structure of ΔRI-NBD1-G3a complex showing that the epitope is located in the periphery of CFTR. **b** Same alignment as (**a**) with the structure 2PT-NBD1-T2a complex, showing a compatible binding of nanobody D12 in between the NBDs. **c** Same alignment as (**a**) with the ΔRI-NBD1-T8 complex where the nanobody overlaps with the TMDs, indicating that binding is not compatible with this conformational state of CFTR

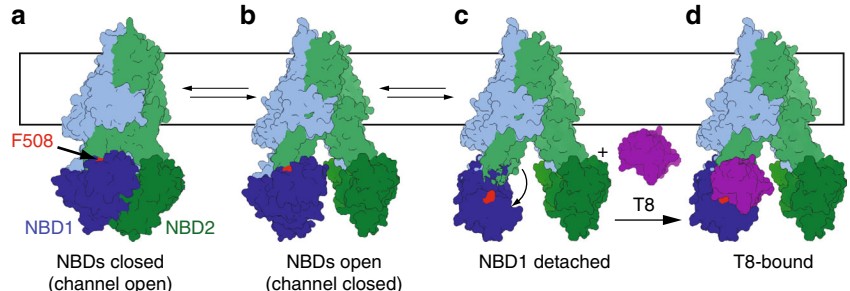

**Fig. 7** NBD1 must undock from the TMDs to allow binding of nanobodies T4 or T8. Current models indicate that CFTR alternates between a state where the two NBDs are in close contact (state **a**), leading to channel opening, and a state where the NBDs separate leading to channel closing (state **b**). State **a** would typically be induced by PKA phosphorylation. States **a** and **b** have been observed by cryo-EM and both bury F508 in the NBD1-TMD interface. Nanobodies T4 and T8 bind an epitope containing F508 (illustrated in red), thus requiring a transient undocking of NBD1 from the TMDs (state **c**). This transient state can be stabilized upon binding of these nanobodies (state **d**)

in the presence of these nanobodies which, upon binding would thus prevent the NBD1-NBD2 interaction required for enzymatic activity. Therefore, while theses nanobodies may be able to stabilize NBD1, they could also hinder channel function. The use of a small molecule mimetic might circumvent such steric limitation, and rational drug design may require carefully taking into account the structures of the different states of CFTR, which are currently emerging.

Superimposing the crystal structures of NBD1-T4 or NBD1-T8 onto the cryo-EM structures of FL-CFTR suggest that these nanobodies should not recognize the full-length protein (Fig. 6c and Supplementary Fig. 6c). Indeed, T4 and T8 completely overlap with the position of the coupling helix of the ICL4, and also with ICL1 and surrounding helices. For ABC proteins in general, ICL4 is considered to be the main interaction site between NBD1 and TMD2, yielding a stable TMD-NBD1 interface. Moreover, while F508 is completely solvent exposed in the isolated NBD1 domain, it becomes completely buried in the NBD1-ICL interface observed in the cryo-EM structures (and thus not available for the nanobodies), while our data have demonstrated that interaction with F508 is strictly required for binding by T4 or T8. Based on these structural data, one would predict that epitope of T4/T8 should not be accessible in FL-CFTR, although our experiments clearly demonstrate that these nanobodies bind mature CFTR, either isolated or in cellular membranes.

Altogether our data would imply that NBD1 must detach from ICL4 and reorient in a manner that allows binding of a ~15-kDa nanobody (schematized in Fig. 7). Interestingly, structural analysis of the interface reveals that the NBD1-TMD interface is significantly weaker than the NBD2-TMD interface, mainly because NBD1 is devoid of usually conserved NBD structural features, namely the S5 β-strand and the h2 α-helix, leading to a reduced interaction surface[3,5]. While this has been previously described as a structural weakness that will render the channel sensitive to modification of the interface (i.e. F508del), it could also be that the reduced interface was evolved to allow undocking of NBD1 for a functional reason. While undocking of NBD1 from ICL4 may appear surprising, it is supported by previous work. Earlier studies have shown that cysteines introduced in NBD1 (at position 508) and ICL4 (at position 1068) which are separated in the cryo-EM structure by about 7 Å ($C_\beta$–$C_\beta$ distance) can be efficiently bridged using crosslinkers of lengths ranging from 4 Å to 24 Å, which could agree with domain motion[34]. Furthermore, crosslinking these two positions with the short reagent 1,1-methanediyl bismethanethiosulfonate (M1M) leads to inhibition of channel gating, which can be reverted by reducing agent, suggesting that a conformational rearrangement of the NBD1-TMD interface may be required for proper function. In addition, HDX experiments on the bacterial ABC homodimeric transporter BmrA have shown that the ICD2 peptide (corresponding to ICL4 in CFTR) exchanges extensively with the solvent, indicating that

is not permanently buried as observed in crystal structure of homologs[35] which suggests that NBD undocking may be happening in other members of the ABC family.

In conclusion, nanobody binding at the NBD1-TMD interface implies that this highly important region is more dynamic than previously appreciated, and therefore suggests the necessity to reconsider how mutations affect the integrity of NBD1 and that of the interface in a physiopathological context. We surmise that the here presented perspective on the dynamics of the interface should have important consequences for therapeutic strategies aimed at modulating its stability.

## Methods

**Human NBD1 expression and purification.** Human ΔRI-NBD1 (residues 387–646, Δ405–436) construct was obtained from Arizona State University Plasmid Repository (clone id: 287374), 2PT-NBD1 mutants (residues 387–646 containing the mutations S492P, A534F, I539T), 2PT-NBD1-RE (2PT-NBD1 with residues 387–678) were constructed using WT-NBD1 construct from ASU (clone id: 287401)[19]. Mutations were introduced by PCR using PfuUltra high-fidelity DNA polymerase from Agilent (catalogue number: 600382) and sequences were confirmed by automated DNA sequencing (UNC-CH Genome Analysis Facility). Proteins were expressed as N-terminal, His$_6$-SUMO fusion proteins in Escherichia coli (BL21(DE3) pLysS cells, Millipore) as described in[5,10,36] with the following modifications. Cells were lysed using a French press and recombinant proteins were purified by nickel ion affinity chromatography (HisTrap HP, 1 ml - GE Healthcare). The His$_6$-SUMO tag was removed using Ulp1 protease at 1/100 weight/weight ratio during 20 min on ice. Then, the cleaved fraction was separated by affinity chromatography (HisTrap HP, GE Healthcare) and further purified by gel filtration on a Superdex 200 10/300 column (GE Healthcare) equilibrated with storage buffer (20 mM Hepes pH 7.5, 150 mM NaCl, 10% (w/v) glycerol, 10% (w/v) ethylene glycol, 2 mM ATP, 3 mM MgCl$_2$, 1 mM Tris(2-carboxyethyl)phosphine (TCEP)). Protein concentration was determined using Coomassie Plus (Bradford) Assay Kit (Thermo Scientific).

**Nanobody cloning and expression and purification.** Nanobodies were cloned in pXAP100 vector as already described[16]. pXAP100 is similar to pMES4 (genbank GQ907248) but contains a C-terminal His$_6$-cMyc tag and allows cloning of the VHH repertoire via SfiI-BstEII restriction sites. Twin-Strep nanobodies were design as follow: the synthetic gene encoding full-length T8 nanobody fused to a C-terminal cleavage site for human rhinovirus 3C (P3C, LEVLFQGP), a cMyc tag (EQKLISEEDL) and a Twin-Strep-tag (WSHPQFEKGGGSGGGSGGGSAWSHPQ-FEK) instead of the His$_6$-cMyc tag was synthesized by Eurofins Genomics and then recloned into pXAP100 vector using NotI/EcoRV restrictions sites. Then, the modified vector was digested with SfiI/NotI to allow insertion of nanobodies T2a, T4, T27, G3a, or D12 in frame with the P3C-cMyc-Twin-Strep sequence. All constructs were verified by sequencing (Eurofins Genomics). Nanobody expression and purification were performed as previously described[16]. Briefly, nanobodies were produced in Escherichia coli (BL21(DE3) pLysS cells, Millipore), purified from the periplasmic extract via either HisPur Ni-NTA resin (ThermoScientific) or Strep-Tactin XT Superflow resin (iba LifeScience) followed by a size exclusion chromatography on a Superdex 200 Increase 10/300 GL (GE Healthcare) equilibrated in 20 mM HEPES pH 7.5, 150 mM NaCl, and 10% (w/v) glycerol.

**NBD1 ELISA assay.** For dose–response assays, Nunc MaxiSorp 96-well plates (ThermoScientific), were coated with 5 µg ml$^{-1}$ NeutrAvidin Biotin-Binding protein (ThermoScientific) overnight at 4 °C and blocked 2 h at room temperature (RT) with 2% milk in phosphate-buffered saline (PBS). Each new reagent addition was preceded by three washes with 200 µl of NBD1 buffer (20 mM HEPES pH 7.5, 150 mM NaCl, and 10% (w/v) glycerol, 10% (w/v) ethylene glycol, 2 mM ATP, 3 mM MgCl$_2$). Then, biotinylated purified NBD1 proteins at 5 µg ml$^{-1}$ were immobilized 30 min at RT followed by 1 h RT incubation with 100 µl various concentrations (0–20 µg ml$^{-1}$) of purified nanobodies. Signal detection was followed using His$_6$-tag specific antibody (Invitrogen, catalogue number: MA1-135, 1:3000 dilution) to detect the nanobodies and secondary antibody anti-mouse coupled to horse radish peroxidase (HRP) (Millipore, catalogue number: AP308P, 1:5000 dilution). 50 µl of 1-Step UltraTMB-ELISA (ThermoScientific) was used as a substrate for the peroxidase and intensity of the reaction was proportional to absorbance measured at 450 nm with SynergyMx (BioTek) after addition of 50 µl H$_2$SO$_4$ at 1 M.

**Thermal shift assay (DSF).** Solutions of either 2PT-NBD1 or F508del-2PT-NBD1 (10 µM final concentration), nanobodies (30 µM final concentration) and 2.5× or 5× concentrated SYPRO Orange Protein Stain (Molecular Probes) diluted in 20 mM HEPES pH 7.5, 150 mM NaCl, 3 mM MgCl$_2$, 2 mM ATP and 10% (w/v) glycerol, 10% (w/v) ethylene glycol, were added to the wells of a 96-well PCR plates type BR white (VWR) in a final volume of 25 µl. Plates were sealed with EasySeal

sheets (Molecular dimensions) and spun 2 min at 900 × g. SYPRO orange fluorescence was monitored in CFX96 Touch Real-Time PCR Detection System (Bio-Rad) using FRET scan mode from 10 to 80 °C in increments of either 1 or 0.2 °C.

**Isothermal titration microcalorimertry (ITC).** Interactions between nanobodies and 2PT-NBD1 was carried out on NanoITC system (TA Instruments) in 0.165 ml cells at 20 °C, 300 r.p.m. syringe stirring. Proteins were extensively dialyzed in 20 mM Hepes buffer pH 7.5, 150 mM NaCl, 10% (w/v) glycerol, 10% (w/v) ethylene glycol, 2 mM ATP and 3 mM MgCl$_2$ for 16 h at 4 °C. Heat of dilution from control experiments of each nanobody titrated into buffer was subtracted from the titration into 2PT-NBD1. Data were integrated analyzed with Origin 7.0 software (OriginLab Corp.).

**Differential scanning calorimetry (DSC).** Calorimetry was performed on the MicroCal VP-Capillary DSC system (Malvern Instruments Ltd). Data were analyzed using the MicroCal Origin software and buffer-buffer heat capacity curve was subtracted from each protein curve. Purified 2PT-NBD1 was incubated with 1.2 molar excess of each nanobody in 20 mM Hepes pH 7.5, 150 mM NaCl, 10% (w/v) glycerol, 10% (w/v) ethylene glycol, 2 mM ATP, 3 mM MgCl$_2$, and incubated for 1 h on ice.

**Crystallization trials and in situ proteolysis.** For each complex formation, nanobodies were SEC purified the day before in 20 mM Hepes pH 7.5, 150 mM NaCl, 10% (w/v) glycerol and mixed with freshly SEC purified NBD1 with 1.2 molar excess of nanobodies, and keeping 2 mM ATP, 3 mM MgCl$_2$ and 1 mM TCEP final concentrations. Protein complexes were incubated 1 h on ice and then concentrated onto 30 kDa MWCO Amicon concentrator (Millipore) until protein concentration reaches 10–18 mg ml$^{-1}$. Proteases from Floppy Choppy kit (Jena Biosciences), either papain or subtilisin A, at a concentration of 1 mg ml$^{-1}$ were added to the purified protein on ice immediately prior to crystallization trials at a ratio of 1 µg protease per 200 µg of protein complex. Crystallization was performed in sitting drops at RT, adding 100 nl of the protease/protein mixture to 100 nl of the precipitant and were set up immediately using Mosquito robot (Art Robbins). For each NBD1-nanobody complex an initial screen of seven commercial screening kits was used (HR-Index, HR-Crystal Screen I&II, MD-Proplex, MD-PACT premier, MD JCSG+, MD-Clear Strategy I, MD-Structure Screen I&II). Crystallization plates were incubated at 20 °C. Single crystals were mounted in CryoLoops (Molecular Dimensions Ltd) and flash-frozen in liquid nitrogen.

**Crystal structure determination.** Native high-resolution X-ray diffraction data were recorded on synchrotron beamline PX2A at SOLEIL in St Aubin, France, with an EIGER X 9 M detector for the ΔRI-NBD1-D12-T4 complex, on beamline i04 at the Diamond Light Source in Didcot, United Kingdom, with a PILATUS 6 M detector for the 2PT-NBD1-T2a-T4 and ΔRI-NBD1-D12-T8 complexes, on beamline i02 at the Diamond Light Source in Didcot, United Kingdom, with a PILATUS 6 M detector for the 2PT-NBD1-T27 complex, and on beamline i24 at the Diamond Light Source in Didcot, United Kingdom, with a PILATUS 6 M detector for the 2PT-NBD1-T27 complex. Data were integrated and scaled using the XDS program[37]. For each NBD1-nanobody complex, the dataset was solved by molecular replacement using Molrep[38]. Subsequently, several cycles of model building, using COOT[39], combined with refinement using BUSTER 2.10.1[40] were conducted. Finally, structure validation was performed with MolProbity[41]. Figures and structural comparisons of the different NBD1-nanobody complexes with the human NBD1 structures previously published (PDB: 2PZE and 2PZF[7], PDB: 2BBO[25], PDB: 1XMJ and 1XMJ[6]) were prepared using UCSF Chimera[42].

**Human CFTR expression and purification.** Two sources of protein were used. A stabilized version of human CFTR protein (stab-CFTR :2PT/ΔRI/R1048A_1172X ) was stably expressed into BHK-21 cells (ATCC; CCL-10) with pNUT vector[43] which were maintained in methotrexate containing medium[44,45] wt-CFTR fused to enhanced green fluorescent protein (His$_{10}$-SUMO*-CFTR$^{FLAG}$-EGFP) was stably expressed in human embryonic kidney (HEK) 293 cell line D165[46] and was PKA phosphorylated with protein kinase A catalytic subunit and affinity purified to homogeneity using NiNTA resin (Qiagen) according to manufacturer-recommended procedures in 50 mM HEPES pH 7.5, 0.15 M NaCl, 10% glycerol, 2.5 mM MgCl$_2$, 2 mM ATP, 0.35 M imidazole, 0.01% Decyl Maltose Neopentyl Glycol (DMNG – Anatrace), 1 mM dithiothreitol. Cells were cultured according to standard mammalian tissue culture protocols including testing for mycoplasma.

**Full-length CFTR ELISA.** Strep-Tactin XT coated microplate (iba Life Science) was coated overnight at 4 °C with Twin-Strep-tagged nanobodies (5 µg ml$^{-1}$). Plate was blocked with 4% milk for 2 h at RT. Then different concentrations of CFTR (10$^{-10}$ to 10$^{-8}$ M) were incubated for 2 h at 4 °C. CFTR binding was detected with monoclonal antibodies L12B4, MM13, 154, 660, 570, 596 specific to CFTR obtained from the CFTR Antibody Distribution Program (http://cftrantibodies.web.unc.edu/available-antibodies, dilution 1: 2000)[47] and then anti-mouse-HRP antibody (Millipore, catalogue number: AP308P, 0.5 µg ml$^{-1}$) for 1 h 30 min at 4 °C. For $B_{max}$ determination Pierce Nickel Coated Plate (ThermoScientific) was coated 1 h

at 4 °C with CFTR (8 µg ml$^{-1}$). Plate was blocked with 4% milk for 2 h at 4 °C. Then different concentrations of nanobodies (10$^{-9}$ to 10$^{-6}$ M) were incubated for 2 h at 4 °C. Nanobody binding was detected with cMyc-tag specific antibody (Sigma, catalogue number: C3956, 0.5 µg ml$^{-1}$) and then anti-rabbit-HRP antibody (Cell Signaling, catalogue number: 7074 S, 1:1000 dilution) for 1 h 30 min at 4 °C. Between each step wells were washed 3 times by aspiration with 50 mM HEPES pH 7.5, 150 mM NaCl, 10% glycerol, 2 mM ATP, 2.5 mM MgCl$_2$, 0.01% DMNG (Anatrace). Incubations were performed in the same buffer with 0.4% milk. Reaction was visualized by using 1-Step Ultra TMB-ELISA (ThermoScientific) and stopped with H$_2$SO$_4$ (500 mM final). Absorbance was measured at 450 nm using SynergyMx (BioTek).

**Flow cytometry**. Parental BHK-21 cells (ATCC; CCL-10) and cells stably over-expressing human wt-CFTR[44,45,48] or 2PT-F508del-CFTR, as described above, were permeabilized with 0.01% n-Dodecyl-β-D-Maltopyranoside (β-DDM - Inalco) at least for 2 h on ice. In the meantime, cells were incubated with 50 µg ml$^{-1}$ nanobodies and DRAQ7 (0.3 µM – Biostatus) to monitor the permeabilization state. Nanobody binding was detected by using His$_6$-tag specific antibody (Invitrogen, catalogue number MA1-135, 1 µg ml$^{-1}$) or cMyc-tag specific antibody (Invitrogen, catalogue number 13-2500, 2 µg ml$^{-1}$) and then anti-mouse-Alexa Fluor 488 (Invitrogen, catalogue number A11001, 1.3 µg ml$^{-1}$) at least for 30 min on ice. Cells were washed one time between each step by centrifugation (200 × g for 5 min at 4 °C). All incubations (100 µl) and washes (1.5 ml) were performed in PBS with 6% fetal bovine serum (FBS) and 0.01% β-DDM on ice. Cells fluorescence was measured with Gallios Flow Cytometer (Beckman Coulter). Data were analyzed with Kaluza software.

**CFTR pull-down**. Human wt-CFTR was extracted from BHK-21 cells pellet by solubilization with 1% DMNG in PBS with proteases inhibitors for 1 h at 4 °C. The cells debris were removed by centrifugation (16,000 × g for 30 min at 4 °C). Supernatant was diluted 10 times in PBS with proteases inhibitors plus 10 mM imidazole and incubated at least for 30 min on HisPur Ni-NTA Resin (Thermo Scientific) pre-loaded with nanobodies. Resin was washed with 40 column volumes of PBS with 300 mM NaCl. Nanobodies were eluted with 200 mM imidazole in PBS. Presence of CFTR in each sample was detected by SDS-PAGE and immunoblotting.

**SDS-PAGE and immunoblotting**. Cell extracts were separated by SDS-PAGE on 7.5% polyacrylamide gel and transferred to nitrocellulose membrane (Bio-Rad) for immunodetection. After blocking for 1 h with 5% bovine serum albumin (BSA) in Tris-buffered saline added 0.05% Tween-20 (TBST), CFTR was detected using monoclonal antibody mAb 596, IgG2b (CFTR Antibody Distribution Program, dilution 1: 2000)[47] for 1 h in blocking buffer. Blot was washed 3 times 5 min and incubated with anti-mouse-HRP antibody (Millipore, catalogue number: AP308P, 0.2 µg ml$^{-1}$) for 1 h in TBST. Membrane was washed 3 times for 5 min. CFTR was visualized by chemiluminescence using Luminata Forte Western HRP Substrate (Millipore) and detected with ImageQuant 400 (GE Healthcare).

**ATPase activity and functional stability assays**. To determine effect of nano-bodies on functional stability, aliquots of purified wt-CFTR (25 nM) were pre-incubated 1 h on ice with 1 µM nanobody (or 15 µM, in the case of G3a). Substrate α-[$^{32}$P]-ATP (2 µl) was then added for measurement of ATPase activity as previously described[49]. Then, nanobody protection against thermal denaturation was determined after a 30 min thermal challenge of the protein complexes followed by an assay of residual ATPase.

**NanoDSF**. Purified stabilized human CFTR (stab-CFTR: 2PT/ΔRI/R1048A_1172 ×) was concentrated to 0.5 mg ml$^{-1}$ and mixed with 0.1 mg ml$^{-1}$ nanobody (~1:2 molar ratio) and capillaries were loaded with a volume of 10 µl. The capillaries were placed into trays of Prometheus NT.48 (Nanotemper) and subjected to the fluorescence analysis. The emission of fluorescent radiation with the wavelengths of 330 nm and 350 nm was measured with the temperature changes from 25 to 85 °C, with the rate of 1 °C min$^{-1}$. The first derivative of 350 nm fluorescence was used to determine the melting temperature of the proteins.

**Statistical analysis**. Affinity constants ($K_D$) and thermodynamic parameters from ITC experiments were determined using one-site binding model with MicroCal Origin 7.0 software (OriginLab Corp.). Dose–response ELISA curves of each Nb binding to either isolated NBD1 or purified FL-CFTR were fitted using the sigmoidal dose–response equation from GraphPad Prism 3. DSC data were analyzed with the MicroCal Origin 7.0 software (OriginLab Corp.), from which the unfolding temperature (Tm) was obtained.

**Reporting Summary**. Further information on research design is available in the Nature Research Reporting Summary linked to this article.

## Data availability

Data supporting the findings of this manuscript are available from the corresponding author upon reasonable request. A reporting summary for this Article is available as a Supplementary Information file. The atomic coordinates and structure factors reported in this paper have been deposited in the Protein Data Bank (PDB). The accession numbers for the structures reported in this paper are PDB: 6GJS, 6GJQ, 6GJU, 6GK4, and 6GKD. The source data underlying Figs. 1a–e, 2b, c, 4a, b, g, 5a, b, e and Supplementary Figs. 3a, b, 4b–d, are provided as a Source Data file.

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

## Acknowledgements

C.G. acknowledges support by the Fond Forton, the Welbio (grant CR-2012S-04R), Vaincre la Mucoviscidose, Mukoviszidose e.V., the Association Luxembourgeoise de Lutte contre la Mucoviscidose, ABCF2, the Chiesi Fondation, the Cystic Fibrosis Foundation and the Fondation ULB. J.R.R. acknowledges support from US grants NIH RO1 DK051870 and CFF RIORDA07XX0. J.S. acknowledges support from Instruct-ERIC, part of the European Strategy Forum on Research Infrastructures (ESFRI), and the Research Foundation Flanders (FWO). CG is a senior Research Associate of the FRS-FNRS. We are grateful to H. Remaut for help for the structural work. We acknowledge Soleil synchrotron (Proxima2A) and Diamond Light Source for time on Beamlines i02, i04 and i24 under Proposals 12718 and 9426.

## Author contributions

Methodology, M.S., M.G., M.O., T.L. and C.G.; Investigation, M.S., M.O., M.G., D.S., A. G.-P., L.H., E.H., E.P., and T.L.; Writing –Original Draft, M.S. and C.G.; Writing –Review & Editing, M.S., M.O., M.G., D.S. A.G.-P., L.H., E.P. T.L., J.S., E.H., I.U., J.R.R. and C.G.; Funding Acquisition, C.G., J.S., J.R.R.; Resources, J.S., I.U. and J.R.R.; Supervision, J.S., I. U., J.R.R. and C.G.

## Additional information

**Competing interests:** A patent application has been filed (application number EP 19171757.8) covering the Nanobodies as well as the structural information of the resolved complexes; inventors are C.G., M.G., M.S., M.O., J.S., E.P., T.L. The other authors declare no competing interests.

