## [Peer Review File · Nature Communications]

Reviewers' Comments:

Reviewer #1:

Remarks to the Author:

CFTR, the ion channel mutated in patients with cystic fibrosis, is a complex and highly regulated protein. The most common disease-associated mutation (F508del) destabilizes the folding of the first nucleotide-binding domain (NBD1) of the channel and its packing against other domains in the protein. Strategies to stabilize NBD1 folding thermodynamically could be therapeutically beneficial.

The authors have developed a panel of nanobodies (Nb) that recognize NBD1 and shown that they stabilize the NBD1 fold to different degrees. Crystallographic structures of the nanobodies reveal three distinct binding epitopes on the WT domain. Some of the nanobodies bind to and stabilize the F508del mutant domain, while others do not. Intriguingly, some Nb also appear to stabilize the full-length CFTR channel, even though the Nb epitopes are not accessible in some or all available structures of different CFTR functional states. This suggests that there may be significant conformational flexibility not reflected in the known structures.

The manuscript presents clearly a wealth of binding and structural data, particularly for complexes formed with the NBD1 domain. The authors note that the Nb binding epitopes could represent targets for therapeutic stabilization and that mimetic compounds could be designed to target them. However, mimetic design is very challenging. It is also well known that CFTR is conformationally adventurous, and the connection of the Nb-binding state(s) of CFTR to physiologically relevant conformations is tenuous. The epitopes "buried" in the known structures may be exposed during conformational excursions to functionally irrelevant states, including conformations that are on-pathway to CFTR degradation. It is also not clear from the experimental evidence that the Nb are binding to the same epitopes on full-length CFTR and in the same orientation as they do on the NBD1. The ability of F508del to disrupt binding of certain Nb to NBD1 and CFTR is consistent with the sharing of an epitope. However, for the Nb that are not disrupted by F508del, identifying disruptive mutation(s) in the NBD1 epitope and showing that they has a similar disruptive effect in CFTR (or vice versa) would help to establish epitope equivalency for the Nb that recognize F508del. In terms of establishing the functional significance of the Nb-stabilized conformations, it would also be important to show that they have an effect (other than inactivation, which could simply reflect a degradation intermediate) on channel electrophysiology. Such evidence would significantly enhance the novelty and relevance of this manuscript for a broader audience.

On a technical note, the R_{free} value reported for 6GKD (0.235) is unusually low for a 3.0Å structure with bond deviations of 0.013 and angle deviations of 1.7 degrees. The lattice contains non-crystallographic symmetry (6 complexes/a.u.?). If the test set reflections were selected randomly, rather than in thin resolution shells, there may have been strong coupling between the working and test sets. Thus, if random selection was used, a new test set should be selected in thin shells and R_{free} should be reset and confirmed, e.g., by use of a simulated annealing protocol. Similar concerns may apply to the other structures that appear to include NCS (6GJQ and 6GK4).

There are a few minor/technical comments:

- The authors should consider using the standardized nomenclature for the mutation lacking Phe508, in place of or in addition to the legacy nomenclature ($\Delta F508$).
- p. 5: The manuscript says Nb interactions are "enthalpy (ΔH) and entropy driven ($T\Delta S$)," but - $T\Delta S$ values for all Nb in Fig. 1e are positive. The manuscript also says that G3a exhibits a "lower free energy factor (ΔG)," although ΔG is actually less favorable, i.e., higher.
- Fig. 4a - are binding data available for a control Nb that doesn't bind CFTR?
- In Table 1, the unit-cell parameters have wrapped in several columns in a way that is inconsistent with the layout shown at the left. Some rows lack units and the space groups are not

formatted according to IUCr conventions (i.e., italics for lattice, subscripts for screw axes).

- there are a few typographical errors (Kd should use italics upper-case K, subscript Roman upper-case D -- throughout; "dephos*pho*phorylated" - p.2; "patient's condition" (singular?) - p. 3; "induce (a) significant shift" -- p. 5; "each of the three epitopes... *were*" - p. 11; "Nanobod*ies* binding was detected" -- p. 26; *32*P (32 superscript) -- p. 27).

Reviewer #2:

Remarks to the Author:

In the presented paper, the authors generated single domain antibodies (nanobodies) against NBD1 of human CFTR. Several crystal structures showing a total of 6 nanobodies that bind to three non-overlapping epitopes were solved. The binders were characterized in diverse biochemical and cellular assays. The binders were found to increase the thermal stability of NBD1 by up to 10°C, by "tying" together different domains of NBD1. Further, they were found to inhibit ATPase activity and were also found to increase stability of full-length CFTR. Interestingly, some binders involving the F506 side chain in their epitopes, which sterically interfere with the coupling helix of ICL4 and thus should only bind the isolated NBD1, but not the full-length transporter due to steric clashes. Yet, the binders were shown recognize full-length CFTR, which suggests that NBD1 can uncouple from the coupling helix and thereby expose the epitope for nanobody binding.

The paper is a nice example of how nanobodies can be used to gain mechanistic insights into a medically relevant membrane protein and (with some reservations that were also recognized by the authors) open avenues how to design small molecules that could stabilize the highly prevalent Delta F506 CFTR mutant. A particular strength of the paper is the large number of structures, resulting in a comprehensive set of binders for mechanistic analyses. Overall, the paper is nicely written and presented and puts the finding into the larger context of CFTR pharmacology. The paper is intelligible for readers outside of the CFTR field (this reviewer is from the transporter/binder field and enjoyed reading the manuscript, which is stand-alone and does not require reading of other papers).

Specific points that need to be addressed by the authors

1) Thermal stabilization by the nanobodies. These assays were conducted using the SYPRO orange method. How can the authors differentiate between the stability effects on NBD1 versus the stability of the added nanobody itself? Unfolding of nanobodies also contributes to SYPRO fluorescence and nanobodies are in general (but not always) quite stable. Further, the nanobody/NBD1 interface is likely to break before the individual proteins are unfolded along the heating ramp. It is in my view mandatory to look at the unfolding curve of the six nanobodies alone, measured at exactly the same nanobody concentration as in the complex with NBD1. Only in this way it can be convincingly shown that the unfolding curves seen for the complexes are not merely the sum of the unfolding curves of the individual proteins (NBD1 + nanobody). A similar issue/problem could also be the case in terms of stabilization of FL-CFTR. In Fig. 5d, the authors show that nanobody T4 stabilizes FL-CFTR. Yet, nanobody T4 can only bind to an uncoupled NBD1 (Fig. 7), which intuitively looks rather destabilized. Of course one may argue that nanobody T4 overcompensates for this destabilization, but I think the evidence needs to be strengthened with regard to this concern.

2) The structural analyses of the nanobody/NBD1 complexes (bottom of p.7, bottom of p.9) are far too detailed. If a reader is really interested in these interfaces, he/she will go to the PISA server (or a similar resource) and look at the interfaced him/herself. For most readers, it likely does not matter whether (to give an example): "The other CDRs also participate in the interface, including a salt bridge observed between D54 in the CDR2 of T8 and K564 of NBD1, and a hydrogen bond observed between D55 of T4 and the backbone...".

3) Nanobodies T4 and T8 were shown to bind to FL-CFTR (Fig. 4a) although binding is expected to be sterically hindered. This raises a number of questions. 1) Is Bmax in the same range as for the other nanobodies that do not sterically interfere? This is important to know, because it is also possible that T4 and T8 recognize a small subset of partially unfolded transporter, i.e. Bmax may be low. 2) Is NBD1 uncoupling reversible? Recovery of ATPase activity after binder wash-off is one

possible way of showing this. 3) What happens in a mutant with a weakened ICL4-NBD1 interface in the context of T4 and T8? I would expect better binding/higher Bmax values. Obviously, this mutation needs to be introduced into ICL4, because T4 and T8 bind much weaker to the NBD1_DeltaF508...In any case, further validation is required to support one of the major findings of the paper, namely the uncoupling of NBD1 from ICL4. Of note, NBD uncoupling had been previously described for BmrA: Mehmood S, Domene C, Forest E, Jault JM. Dynamics of a bacterial multidrug ABC transporter in the inward- and outward-facing conformations. Proc Natl Acad Sci US A. 2012 Jul 3;109(27):10832-6. This work deserves to be mentioned in the discussion.

4) Data of Figure 4. In these flow cytometry assays, cells were permeabilized to get nanobodies into the cytosol. A control nanobody not binding to the target served as control. Given the fact that the differences between control and binders are not huge: how can the authors be sure whether the control nanobody had the same degree of labelling and that the entered concentration was equal to the other nanobodies? To further strengthen the experiment, I suggest to repeat the experiment with a cell line expressing the FL-CFTR_DeltaF508 mutant along with cells expressing the wt cells. In this case one would expect a drop of binding for the T4/T8 nanobodies, that bind much weaker to FL-CFTR_DeltaF508. In this way, one could overcome this concern (differential labelling/concentration differences) because one and the same labelled T4/T8 nanobody at exactly the same concentration could be used.

Minor points

- 1) I suggest to number the six nanobodies from #1-6. E.g. by stating Nb_CFTR#1, #2, etc. or similar. This would highly increase readability of the manuscript. #1-3 recognize the first epitope, #4-5 to the second and #6 for the third. As simple as that.
- 2) Fig. 1e: why is in the table an pEC50 given, and not EC50 as in the text. EC50ies (in nM) would be much easier to read/understand.
- 3) Fig. 1e: Why was ITC not performed with the DeltaF508 NBD1 mutant?
- 4) Middle of p.7: Is there a pattern regarding the interacting framework residues of these three nanobodies versus the other nanobodies of the study?
- 5) Fig. S4: I suggest including the data with the normal NBD1 variant side-by-side with 2PT-NBD1-RE to appreciate the differences.
- 6) Fig. 5a lacks positive control (i.e. conditions in which 100 % activity were determined). Was it buffer or an unrelated binder?
- 7) Superimpositions of nanobodies with FL-CFTR. Have the authors also superimposed the nanobodies with outward-facing FL-CFTR having closed NBDs? I specifically wonder about G3a, which is not inhibitory. Does it really not sterically clash with closed NBDs?

Reviewer #3:

Remarks to the Author:

Summary:

The manuscript by Sigoillot et al. is well written and provides new insights into how NBD1 may be stabilized through nanobody binding. Atomic detail of the binding interfaces between the stabilizing nanobodies and isolated NBD1 provide valuable information on target protein surfaces in NBD1 for the development of further stabilizing small molecule therapeutics. The manuscript ends by proposing a novel conformational dynamic of NBD1 in the full-length protein, in which NBD1 'undocks' from TMD1.

Comments:

- The final proposed model in which NBD1 'undocks' from TMD1 (Fig. 7) is speculative, as it is inferred mostly from data involving isolated NBD1, crystalized with nanobodies in the absence of the full-length protein. The reader is left wondering if the nanobodies T4 and T8 could bind a different location on the full-length protein that would not require 'undocking'. Could the conformational dynamics that enable nanobody binding occur on immature CFTR during trafficking, rather than on mature CFTR? Further, the biological relevance of such conformational dynamics is

unclear, particularly since T4 and T8 nanobody stabilization of CFTR is measured for a detergent solubilized state and in semi-permeabilized cells. More conclusive evidence for such large-scale motions in full-length CFTR should be provided (e.g. cross-linking, mutagenesis, or structural studies).

- The main findings in the paper appear to be the identification of specific regions of NBD1 that could contribute to stabilization of the full-length protein if bound by nanobodies, peptide mimetics, or small molecules. As such, the title of the paper, citing "conformational dynamics", is perhaps misleading.

- Additionally, there are areas in the text where data for some nanobodies is unexplainably missing. For example, Fig. 4 panel f, immunoblots of full-length CFTR pull-downs are not provided for nanobodies D12, T27, and T4. Fig. 5, panel a and b, ATPase activity is missing for full-length CFTR and T4 nanobody. Although the nanobodies T4 and T8 are appreciably similar, data should be provided for all nanobodies.

- While the main findings of the manuscript are interesting and will benefit the CFTR field, the conclusions in the present version of the manuscript seem ultimately too speculative for publication in Nature Communications.

Reviewer #4:

Remarks to the Author:

Manuscript background information

In this manuscript, the authors describe work leading to the identification and characterization of a series of nanobodies from llamas, developed by phage display screening and selection. Antigen used for immunization, importantly, was a version of wildtype nucleotide-binding domain 1 (NBD1) of CFTR that included a series of mutations shown by this group and others to introduce thermal stability. In characterizing the six selected nanobodies, the authors used a variety of biophysical techniques and compared the binding to the same wildtype NBD1 protein and to NBD1 bearing the most common CF-causing mutation, F508del; importantly, all of these proteins still included the stabilizing mutations, meaning that binding data are not necessarily reflective of the truly wildtype target. The authors then also showed that several of the nanobodies bind to full-length CFTR. First, they used purified full-length CFTR in ELISA assays, and then they used full-length CFTR in situ in BHK cells permeabilized to enable access of the nanobodies to their intracellular target sites. However, the reader is not told whether the versions of full-length CFTR used in these latter experiments also includes the stabilizing mutations.

The experiments are nicely done (indeed, the ITC data are super-clean), and the paper is reasonably well-written with the caveat that the authors overstate the study's importance. A few very important experiments still need to be done.

Comments for transmission to the authors:

The authors have accomplished some very important and very interesting research as reported in this manuscript. However, they are quite ambitious in the stated implications of their research. Claiming that a small-molecule derivative of these nanobodies could be a therapeutic without showing any physiological correction of trafficking of mutant CFTR, and while also showing a significant decrease in ATPase activity when the nanobodies interact with the protein, is misleading and contrary to the potential of these tools as therapeutics. Also, claiming that NBD1 likely moves away from the TMDs based on a superimposed version of the NBD1-nanobody crystal structure on the full-length crystal structure is premature.

Most of the data are technically sound.

In general, the authors do not do a good job of explaining the very surprising results regarding the availability of the apparent binding sites in the full-length protein. There has been quite a lot of discussion in the CFTR community regarding the question of whether the two NBDs fully disengage EVER in the physiological context. Not only does their proposed model (Figure 7) suggest that the NBDs do fully disengage, by a degree large enough for the D12/T2a/T27 nanobodies to reach their binding sites, but then for the T4/T8 nanobodies to reach their binding sites the model requires the NBDs to disengage fully then followed by a considerable degree of unfolding of NBD1. This is a really hard sell, and therefore the bar should be set a good bit higher for us to believe this interpretation. Certainly, more experiments with full-length CFTR studied under physiological conditions are required in order to support these conclusions.

Major comments:

1. Figure 1. It is really disturbing how often the authors end their studies at $n=2$. To calculate standard deviation with $n=2$ is a huge stretch.
2. The authors never address the question of whether NBD1 peptides used as immunogens, or in the biophysical assays upon which the paper is based, remain monomeric. If the peptides are forming dimers, interpretations would be greatly changed. This also leads one to question why all of the biophysical assays were done with only NBD1 instead of with NBD1 plus NBD2 peptides, which would be more physiologically relevant.
3. Similarly, NBD1 almost certainly has nucleotide bound most of the time, given cellular conditions. Why were biophysical studies done without ATP or a non-hydrolyzable analog?
4. To achieve a crystal structure for some of their constructs the authors had to treat their samples with proteases. This reviewer is nervous about that, as that is a pretty significant change to the NBD1 structure, even if the changes were all far from the binding site.
5. The authors need to add a supplemental figure confirming that the negative control nanobody actually does not bind to CFTR.
6. At the end of the section "Interaction of nanobodies with full-length CFTR," the authors note that the nanobodies cause a drastic decrease in ATPase activity. These data would suggest that these nanobodies (or some small-molecule derivative based upon them) would be bad candidates for therapeutics. Hence, CFTR ion channel activity (WT and F508del) should be studied in the presence of these nanobodies. For purified protein, the Riordan lab has strengths in bilayer reconstitution studies. For the cellular context, excised patch recordings can be used to prevent the need for permeabilizing the cell membrane. It is very surprising that there were no functional studies included beyond ATPase assays.
7. In this same section, the authors showed in Figure 4a that nanobodies T2a, T8, and G3a bind purified CFTR protein with high affinity. Did they not test the other three nanobodies in similar experiments? Seems odd that these data were not provided.
8. Importantly, the authors need to test the ability of the nanobodies to bind full-length CFTR under more physiological conditions. In the flow cytometry experiments described in Figure 4, the cells have been permeabilized (to enable access of the nanobodies to their cytoplasmic binding sites) and the cytoplasm is flushed with PBS plus detergent. We have no idea how this will affect the conformation of CFTR. Given the very surprising results presented in this paper, as noted above, this experiment needs to be repeated under physiological conditions that control the state of CFTR such as: inclusion of divalent cations, inclusion of ATP, phosphorylation of CFTR.
9. Page 9, paragraph 3. The authors found that the interaction of nanobodies D12, T2a, and T27 with NBD1 overlaps with the location of the regulatory extension (RE), although the RE was removed from the NBD1 construct used for immunization. When the authors tested the ability of these three nanobodies to bind NBD1 with the RE, the wording used to describe the result is misleading. Here, the authors state that they "observe a high-affinity binding... albeit with a decrease in apparent EC_{50} ," in regards to the nanobodies binding 2PT-NBD1-RE. This sentence is slightly misleading, as a decrease in binding EC_{50} is directly caused by a decrease in binding affinity. Therefore, it would be more accurate to say that they "observe a decreased but still high binding affinity."

10. The authors claim that these nanobodies stabilize F508del, and therefore likely increase trafficking, but never treat CFTR-expressing cells and assess the change in cell surface expression of CFTR. This is vital if the authors want to show that these nanobodies increase trafficking and might actually be informative toward therapeutic development. However, the authors would have to somehow permeabilize the cells to allow the nanobody in to reach their binding sites, or use another delivery method. Certainly, this will be difficult, but needs to be explored.

11. There are odd cases of missing results. For example, why are there no data in Figure 5 for ATPase activity or ΔT_m for the T4 nanobody? Why are there no data for ITC on the D12 nanobody, especially since D12 appears in 3 of your 4 double nanobody crystallization conditions? Why is there no data for nanoDSF on the G3a or D12? Including those two, considering they are on the ends of the thermal shift data presented in the earlier panel (Figure 5b), would support the findings in both methods.

12. Why does the NBD1+T4 have two thermal shift peaks in Figure 2? Is this partial unfolding of the nanobody or NBD1; and why only for that nanobody? The authors need to address this experimental concern.

13. The authors mention that papain and subtilisin A were used to generate diffracting crystals, but that the nanobody binding interface with NBD1 was unaffected. Please provide a structural comparison figure to support this evidence.

14. On pg 17, in the Conclusions, the authors make a pretty large logical jump saying that NBD1 probably springs out from the TMD and then comes back to allow T4/T8 nanobodies to bind. However, this is based on the NBD1-alone crystal structure superimposed on the full-length structure. It is completely possible that the nanobody can force its way in between NBD1 and TMD, or capture a conformation rarely experienced due to the high affinity of the nanobody, thereby creating the false impression of a new conformation. It is also possible that the nanobody does not bind in exactly the same way when the rest of the protein is there. These hypotheses are especially possible since the change in T_m is less when T4/T8 is added to FL-CFTR as compared to NBD1 alone. The change in conformation schematically proposed in Figure 7, to accommodate the generation of a binding site for the T4 nanobody, would likely be energetically very expensive. The authors should include a consideration of how much energy it would take to unfold NBD1 to this extent.

15. The authors talk about how large the binding interface of these nanobodies is (600-1000 angstroms), yet claim a small molecule could be developed to replicate nanobody binding. If this is going to be a claim, the authors need to identify a small section of the binding interface that is vital for stability. This, however, would basically require the development of the proposed small molecule. The binding interface for nanobodies is considerably larger than that of small molecules.

Minor comments:

1. Page 2: The authors misstate the data on number of mutations. Certainly, >2000 mutations in the CFTR gene have been identified in the human population. However, it is not known what number of these are actually disease-causing mutations. The authors are referred to the CFTR2 database for clarification: <https://www.cftr2.org/>

2. Page 10. In the section "Interaction of nanobodies with full-length CFTR," when the authors reference "mature FL-CFTR," it is important for them to explain what that means in relation to immature FL-CFTR, cellular localization, and apparent size on electrophoresis. Not all readers of Nature Communications will be familiar with "mature" as a description.

3. It is important for the authors to comment on the lack of immunogenicity of nanobodies, as the reader may become confused as to how a biologic could be used as long-term therapy without mounting a huge immune response.

4. It is not optimal to use thermostabilized mutant CFTR for thermostability assays, but it is understandable. It would be wonderful to see these experiments repeated in truly WT constructs.

5. Can the authors confirm that no crystals formed for the nanobody D12 or T4 alone, or no good diffracting crystals, or was this not even tested.

6. Please include the original ΔRE -NBD1 nanobody binding data (EC_{50}) in a table with the RE-NBD1 construct nanobody binding data in Supplementary Figure 4 to confirm no change in binding.

7. Please include the experimental condition of the "CTRL" in figure legend for Figure 4f.
8. Please include a bar for the WT-FL-CFTR ATPase activity data (100%)
9. Maintain the color code from Figure 4a-d in Figure 5.
10. Please provide more explanation of the purification of the thermostabilized human CFTR construct from BHK-21 cells (at least information on whether the construct is tagged or untagged).
11. Typos and errors throughout the manuscript
 - a. I suggest that the authors should have the manuscript reviewed for proper English. There are a great many small errors that would likely not be made by an author whose first language is English.
 - b. Include both thermostabilized and wt-NDB1. Page 3.
 - c. Table 1, please include the word supplemental table 1. Page 6.
 - d. Incomplete sentence/thought at the end of the section "Structures of nanobody-NBD1 complexes". Page 8.
 - e. Van der Waals. Page 10.
 - f. Use a delta (Δ) symbol in supplemental table 1 instead of using a D to represent deletion; it's currently confusing as there are nanobodies using that nomenclature (D12). Page 33.
 - g. Supplementary figure 5. Wrong labeling for all subparts (a, b, and c), do not match the main text and figure legend.

CFTR, the ion channel mutated in patients with cystic fibrosis, is a complex and highly regulated protein. The most common disease-associated mutation (F508del) destabilizes the folding of the first nucleotide-binding domain (NBD1) of the channel and its packing against other domains in the protein. Strategies to stabilize NBD1 folding thermodynamically could be therapeutically beneficial.

The authors have developed a panel of nanobodies (Nb) that recognize NBD1 and shown that they stabilize the NBD1 fold to different degrees. Crystallographic structures of the nanobodies reveal three distinct binding epitopes on the WT domain. Some of the nanobodies bind to and stabilize the F508del mutant domain, while others do not. Intriguingly, some Nb also appear to stabilize the full-length CFTR channel, even though the Nb epitopes are not accessible in some or all available structures of different CFTR functional states. This suggests that there may be significant conformational flexibility not reflected in the known structures.

The manuscript presents clearly a wealth of binding and structural data, particularly for complexes formed with the NBD1 domain. The authors note that the Nb binding epitopes could represent targets for therapeutic stabilization and that mimetic compounds could be designed to target them. However, mimetic design is very challenging. It is also well known that CFTR is conformationally adventurous, and the connection of the Nb-binding state(s) of CFTR to physiologically relevant conformations is tenuous.

The epitopes "buried" in the known structures may be exposed during conformational excursions to functionally irrelevant states, including conformations that are on-pathway to CFTR degradation.

We agree with the reviewer that CFTR conformation(s) which permit T4/T8 binding could be transient and/or non-functional. However, they may still be very much biologically relevant: if on-pathway to unfolding and/or degradation, they may be involved in pathogenesis. We note however that T4/T8 binding strongly stabilizes full-length CFTR which would argue against unfolding.

We would like to mention that a new experiment (requested by referee #2) indicates that the number of binding sites accessible to T8 is lower than for T2a suggesting that only a subpopulation (albeit significant) of CFTR is exposing this epitope (Fig. 4b).

It is also not clear from the experimental evidence that the Nb are binding to the same epitopes on full-length CFTR and in the same orientation as they do on the NBD1.

This point was raised by several reviewers, here is our answer:

We are not aware of a single case where a given monoclonal antibody/nanobody would recognize different epitopes (or bind differently) two forms of the same proteins (i.e. domain vs. full-length). As we observe high affinity binding to both isolated NBD1 and full-length CFTR, it seems improbable that the nanobodies would bind differently the two forms. Our high-resolution structures illustrate how the given sequences of each nanobody achieve detailed interactions with specific sites of NBD1 and it expected that these interactions will be achieved in both cases. We observed such behaviour for both T4 and T8, with high affinity binding on both NBD1 and full-length CFTR although the two nanobodies bear different sequences. It thus seems that the unexpected/unlucky scenario of two epitopes is quite unlikely.

However, in order to answer the reviewers' request, we have performed a new experiment to demonstrate that the nanobodies behave identically on isolated NBD1 and full-length CFTR, by taking advantage of the fact that the epitope of T8 requires F508 and that T8 is unable to bind 2PT-F508del-NBD1. In order to solely look at epitope recognition and not protein unfolding, we used an F508del mutant full-length protein stabilized by the 2PT mutations (2PT-F508del-CFTR). This construct is known to mature and properly channel ions (Aleksandrov JMB 2012). Flow cytometry experiments were performed with representative nanobodies (T2a, T8 and G3a). As shown below, T2a and G3a are able to recognize the protein, demonstrating that the overall fold of NBD1 is indeed native in this construct. However, T8 is unable to recognize 2PT-F508del-CFTR, thus demonstrating that F508 is indeed key

part of the T8 epitope in the context of full-length CFTR. In our opinion, this demonstrates that the epitopes are indeed similar in isolated NBD1 and in the complete channel.

Left panel: Flow cytometry analysis of nanobodies T2a, T8 and G3a on BHK-21 cells expressing 2PT-F508del showing increased labelling for T2a and G3a, but not T8.

Right panel: Quantification of data illustrated in (h). Average of 3 independent experiments (\pm SEM).

The data are now presented in Figure 4.

The ability of F508del to disrupt binding of certain Nb to NBD1 and CFTR is consistent with the sharing of an epitope. However, for the Nb that are not disrupted by F508del, identifying disruptive mutation(s) in the NBD1 epitope and showing that they has a similar disruptive effect in CFTR (or vice versa) would help to establish epitope equivalency for the Nb that recognize F508del.

As described above we believe that our flow cytometry data on the 2PT-F508del-CFTR demonstrate that the epitopes are the same in isolated domain and full-length protein.

In terms of establishing the functional significance of the Nb-stabilized conformations, it would also be important to show that they have an effect (other than inactivation, which could simply reflect a degradation intermediate) on channel electrophysiology. Such evidence would significantly enhance the novelty and relevance of this manuscript for a broader audience.

The possible effect of nanobodies on CFTR function is partially addressed in our manuscript as we show that several of them affect ATPase activity (Fig. 5a). Understanding if the nanobodies can modulate channel activity could be important to define them as research tools, and experiments are currently planned. However, we do not feel that the present study requires such challenging characterization to demonstrate that specific stabilization of CFTR can be achieved and that new conformational states are uncovered. Obviously stabilizing molecules (whether nanobody-inspired or not), will need to favour channel opening not repress it, but as stated in our manuscript, the current set of nanobodies are not intended as direct therapeutic candidate, but rather provide the structural template to guide the design of CFTR stabilizers.

On a technical note, the Rfree value reported for 6GKD (0.235) is unusually low for a 3.0Å structure with bond deviations of 0.013 and angle deviations of 1.7 degrees. The lattice contains non-crystallographic symmetry (6 complexes/a.u.?). If the test set reflections were selected randomly, rather than in thin resolution shells, there may have been strong coupling between the working and test sets. Thus, if random selection was used, a new test set should be selected in thin shells and Rfree should be reset and confirmed, e.g., by use of a simulated annealing protocol. Similar concerns may apply to the other structures that appear to include NCS (6GJQ and 6GK4).

The reviewer is correct that high NCS may lead to unwanted coupling between the working and the test sets, as discussed by Kleygvet back in 1996 who suggested the use of thin resolution shells to circumvent this issue. We also note that the presence of NCS will increase the number of constrains used by the refinement programs thus leading to better refinements, especially at medium resolution where the number of observables is limited. The

refinement program that we have used here (BUSTER-TNT) is known in the community to perform particularly well in such cases. And thus the apparent low R_{free} was not surprising to us.

Nevertheless, we performed the verification suggested by the reviewer for 6GKD. Namely, we used the "Simple Dynamics" procedure implemented in the Phenix package to "shake" the coordinates and deviate from the refined model. Then we replaced the `FreeR_flags` originally used with a new set selected using thin resolution shells, using the SFTOOLS (from the ccp4 suite) with the "SHELL" parameter.

Using BUSTER we then performed 10 refinement cycles as previously done for 6GKD (using the "autoncs" macro which incorporates NCS restraints in the procedure). The resulting R_{free} is 0.232 (as calculated by the PDB validation tool), thus identical to the one calculated for the deposited structure. Therefore, we are confident that the low R values of 6GKD (and the other structures) reflect the intrinsic quality of the refined model rather than a bias in the test set used.

There are a few minor/technical comments:

- The authors should consider using the standardized nomenclature for the mutation lacking Phe508, in place of or in addition to the legacy nomenclature ($\Delta F508$).

We have now used the F508del nomenclature.

- p. 5: The manuscript says Nb interactions are "enthalpy (ΔH) and entropy driven ($T\Delta S$)," but $-T\Delta S$ values for all Nb in Fig. 1e are positive. The manuscript also says that G3a exhibits a "lower free energy factor (ΔG)," although ΔG is actually less favorable, i.e., higher.

The sentences have been corrected as follows:

The binding signature of each nanobody revealed that all the interactions are enthalpy (ΔH) driven. Note that nanobody G3a binding to 2PT-NBD1 exhibits less negative free energy factor (ΔG) explaining its reduced affinity (Fig. 1e and Supplementary Fig. 2).

- Fig. 4a - are binding data available for a control Nb that doesn't bind CFTR?

We have always used a non-CFTR nanobody control. For clarity, the data is now presented with the signal from the control nanobody subtracted from the signal of the CFTR-specific nanobodies.

- In Table 1, the unit-cell parameters have wrapped in several columns in a way that is inconsistent with the layout shown at the left. Some rows lack units and the space groups are not formatted according to IUCr conventions (i.e., italics for lattice, subscripts for screw axes).

Table 1 has been corrected and formatted according to IUCr conventions.

- there are a few typographical errors (K_d should use italics upper-case K, subscript Roman upper-case D

-- throughout; "dephosphorylated" - p.2; "patient's condition" (singular?) - p. 3; "induce (a) significant shift" -- p. 5; "each of the three epitopes... *were*" - p. 11; "Nanobodies* binding was detected" -- p. 26; ³²P (32 superscript) -- p. 27).

Typographical errors have been corrected.

In the presented paper, the authors generated single domain antibodies (nanobodies) against NBD1 of human CFTR. Several crystal structures showing a total of 6 nanobodies that bind to three non-overlapping epitopes were solved. The binders were characterized in diverse biochemical and cellular assays. The binders were found to increase the thermal stability of NBD1 by up to 10°C, by “tying” together different domains of NBD1. Further, they were found to inhibit ATPase activity and were also found to increase stability of full-length CFTR. Interestingly, some binders involving the F506 side chain in their epitopes, which sterically interfere with the coupling helix of ICL4 and thus should only bind the isolated NBD1, but not the full-length transporter due to steric clashes. Yet, the binders were shown recognize full-length CFTR, which suggests that NBD1 can uncouple from the coupling helix and thereby expose the epitope for nanobody binding.

The paper is a nice example of how nanobodies can be used to gain mechanistic insights into a medically relevant membrane protein and (with some reservations that were also recognized by the authors) open avenues how to design small molecules that could stabilize the highly prevalent Delta F506 CFTR mutant. A particular strength of the paper is the large number of structures, resulting in a comprehensive set of binders for mechanistic analyses. Overall, the paper is nicely written and presented and puts the finding into the larger context of CFTR pharmacology. The paper is intelligible for readers outside of the CFTR field (this reviewer is from the transporter/binder field and enjoyed reading the manuscript, which is stand-alone and does not require reading of other papers).

Specific points that need to be addressed by the authors

1) Thermal stabilization by the nanobodies. These assays were conducted using the SYPRO orange method. How can the authors differentiate between the stability effects on NBD1 versus the stability of the added nanobody itself? Unfolding of nanobodies also contributes to SYPRO fluorescence and nanobodies are in general (but not always) quite stable. Further, the nanobody/NBD1 interface is likely to break before the individual proteins are unfolded along the heating ramp. It is in my view mandatory to look at the unfolding curve of the six nanobodies alone, measured at exactly the same nanobody concentration as in the complex with NBD1. Only in this way it can be convincingly shown that the unfolding curves seen for the complexes are not merely the sum of the unfolding curves of the individual proteins (NBD1 + nanobody).

We had not provided in the original manuscript the unfolding curves of the isolated nanobodies to avoid overloading of data. The thermofluor assay is well suited here because the signal from NBD1 is strong, while the signal from each nanobody range from moderate (for example signal for T4 is visible on Figure 2a) to very weak (for example D12), and is always at higher temperatures than the transition of NBD1. Thus the temperature shifts presented do indeed represent the signal coming from SYPRO binding to NBD1. This is also confirmed by the DSC data (Supplementary Fig. 3). However, as requested by the reviewer, we now present the individual unfolding curves and a table reporting melting temperature of each nanobody alone and in presence of NBD1 protein in the Supplementary Figure 3 b,c.

Please note that the thermal-shift conditions were optimized for NBD1 and not to test nanobody stability. This assay is protein dependent and optimization of dye and/or protein concentrations may be necessary for optimal assay performance. For example, nanobodies T8 and G3a exhibited a very low fluorescence signal in such conditions. In order to accurately determine melting temperature of all the individual nanobodies we have performed additional experiments at higher concentration of SYPRO Orange dye (5x instead of 2.5x excess). As presented in the new table, we were able to accurately determine T_m for T8 and G3a, and did not observed significant change for the other nanobodies (Supplementary Fig. 3b).

A similar issue/problem could also be the case in terms of stabilization of FL-CFTR. In Fig. 5d, the authors show that nanobody T4 stabilizes FL-CFTR. Yet, nanobody T4 can only bind to an uncoupled NBD1 (Fig. 7), which intuitively looks rather destabilized. Of course one may argue that nanobody T4 overcompensates for this destabilization, but I think the evidence needs to be strengthened with regard to this concern.

Our data does not suggest that the uncoupled state is destabilized. As shown in Figure 5 the signal for FL-CFTR and T4 are clearly distinct. Furthermore, the functional T_m shown in Figure 5b can only originate from CFTR, and there

also a clear stabilization effect is seen, which is entirely consistent with the unfolding data in Figure 5c,d. As such we strongly believe that the data show a specific stabilization of CFTR and is not an artefact of convoluted signals.

2) The structural analyses of the nanobody/NBD1 complexes (bottom of p.7, bottom of p.9) are far too detailed. If a reader is really interested in these interfaces, he/she will go to the PISA server (or a similar resource) and look at the interfaced him/herself. For most readers, it likely does not matter whether (to give an example): "The other CDRs also participate in the interface, including a salt bridge observed between D54 in the CDR2 of T8 and K564 of NBD1, and a hydrogen bond observed between D55 of T4 and the backbone...".

We agree that the appropriate depth of a structural description is always difficult to establish. Here we are providing some details to illustrate the specificities of the interactions required to form the complexes. We think this is important to illustrate how the sequences have been matured to match the molecular epitopes. We think this is good to have in mind when then considering the interactions with the full-length protein.

3) Nanobodies T4 and T8 were shown to bind to FL-CFTR (Fig. 4a) although binding is expected to be sterically hindered. This raises a number of questions. 1) Is B_{max} in the same range as for the other nanobodies that do not sterically interfere?

We thank the reviewer for this suggestion. We have performed and compared ELISA experiments on immobilized (solubilized) CFTR or immobilized nanobodies with our three representative nanobodies. The result is striking (see below). While on isolated NBD1, where each epitope are expected to be equally accessible; all three nanobodies reach similar plateaus, indicating that, indeed the numbers of binding sites are equivalent. In contrast, when tested against immobilized full-length CFTR, we observed that on average ($N=3$), the B_{max} for T8 is about 36% that of T2a (used as reference), indicating that at most 1/3 of the CFTR population is in the uncoupled state. A surprising result is that G3a also shows reduced B_{max} although its epitope appears entirely accessible in the cryo-EM structure of human CFTR. As this nanobody recognizes the SDR (structurally-diverse region), it is likely due to diversity in the conformational states in the context of the entire protein.

(a) Dose-response ELISA of interactions between WT-CFTR and nanobodies. Immobilized Twin-Strep-tagged nanobodies (T2a, T8 and G3a) were incubated with different concentrations of purified human CFTR. Data were normalized to maximal response of T2a. (b) Purified WT-CFTR was immobilized on His-grab plates and incubated with increasing concentrations of nanobodies. For both (a) and (b) Graph depicts one representative of at least three independent experiments. Error bars represent standard deviations of triplicates. (c) Average B_{max} values calculated for curves in panel B. Average of at least 3 independent experiments (\pm SEM).

2) Is NBD1 uncoupling reversible? Recovery of ATPase activity after binder wash-off is one possible way of showing this.

The reviewer's question is sensible, but unfortunately a simple wash-off of high affinity nanobody has been shown to be difficult to achieve and thus may not allow to properly answer the question. However, single-molecule FRET experiments should be able to follow individual uncoupling/recoupling

events. While these experiments are currently under development, there are technically very challenging and will be the center of a separate study.

3) What happens in a mutant with a weakened ICL4-NBD1 interface in the context of T4 and T8? I would expect better binding/higher Bmax values. Obviously, this mutation needs to be introduced into ICL4, because T4 and T8 bind much weaker to the NBD1_DeltaF508...In any case, further validation is required

The reviewer's suggestion is sensible, but also extremely challenging, as identifying and characterizing weakening mutations without deleterious effects on the protein/epitopes may be particularly difficult. We believe that, in the context of a communication paper, the new experiments, now integrated with the literature (see below and in the Discussion section), provide strong support to our hypothesis.

to support one of the major findings of the paper, namely the uncoupling of NBD1 from ICL4. Of note, NBD uncoupling had been previously described for BmrA: Mehmood S, Domene C, Forest E, Jault JM. Dynamics of a bacterial multidrug ABC transporter in the inward- and outward-facing conformations. Proc Natl Acad Sci US A. 2012 Jul 3;109(27):10832-6. This work deserves to be mentioned in the discussion.

We are grateful to the reviewer for pointing out this paper to us, which also led us to revisit older crosslinking data on CFTR in the context of the cryo-EM structure. This provides stronger support to the undocking hypothesis, as described in the added paragraph (see Discussion):

While undocking of NBD1 from ICL4 may appear surprising, it is supported by previous work. Earlier studies have shown that cysteines introduced in NBD1 (at position 508) and ICL4 (at position 1068) which are separated in the cryo-EM structure by about 7 Å (C_β-C_β distance) can be efficiently bridged using crosslinkers of lengths ranging from 4 Å to 24 Å, indeed supporting the concept of large domain motion (Serohijos, PNAS 2008). Furthermore, crosslinking these two positions with the short M1M leads to inhibition of channel gating, which can be reverted by reducing agent, suggesting that such domain motion may even be required for proper function. In addition, HDX experiments on the bacterial ABC homodimeric transporter BmrA have shown that the ICD2 peptide (corresponding to ICL4 in CFTR) exchanges extensively with the solvent, indicating that is not permanently buried as observed in crystal structure of homologs (Mehmood PNAS 2012) which suggests that NBD undocking may be happening in other members of the ABC family.

4) Data of Figure 4. In these flow cytometry assays, cells were permeabilized to get nanobodies into the cytosol. A control nanobody not binding to the target served as control. Given the fact that the differences between control and binders are not huge: how can the authors be sure whether the control nanobody had the same degree of labelling and that the entered concentration was equal to the other nanobodies? To further strengthen the experiment, I suggest to repeat the experiment with a cell line expressing the FL-CFTR_DeltaF508 mutant along with cells expressing the wt cells. In this case one would expect a drop of binding for the T4/T8 nanobodies that bind much weaker to FL-CFTR_DeltaF508. In this way, one could overcome this concern (differential labelling/concentration differences) because one and the same labelled T4/T8 nanobody at exactly the same concentration could be used.

We understand the reviewer's concern and should stress that all flow cytometry experiments were fully controlled with labelling of parental cells, using identical concentration of nanobodies. We now have added this control which show not difference in labelling (Figure 4d).

Furthermore, we have also now performed a new experiment using, as suggested F508del-CFTR (Figure 4h). However, in order to only look at epitope modification and not global protein unfolding, we have used a stabilized version (2PT-F508del-CFTR).

Under these conditions, nanobodies T2a and G3a recognize efficiently the protein, indicating that the NBD1 domain is properly folded. However, in spite the forced stabilization, the T8 nanobody is not able to recognize this protein which misses one key interacting residue in the interface (F508). We conclude

that the labelling differences are indeed due to changes in binding mode and that the epitopes recognized on the full-length protein are the one characterized by crystallography on isolated NBD1.

Minor

points

1) I suggest to number the six nanobodies from #1-6. E.g. by stating Nb_CFTR#1, #2, etc. or similar. This would highly increase readability of the manuscript. #1-3 recognize the first epitope, #4-5 to the second and #6 for the third. As simple as that.

The reviewer's suggestion is sensible but the current naming is, as often the case, historical. As these nanobodies have now been distributed widely in the CF community, we believe that changing the names would lead to confusion.

2) Fig. 1e: why is in the table an pEC50 given, and not EC50 as in the text. EC50ies (in nM) would be much easier to read/understand.

While nM concentrations appear more "intuitive", the issue comes when performing statistics. All curves are calculated based on a logarithmic concentration range. Thus EC50 distributions are typically not normal (not symmetrical in fact). However, logEC50 have distributions much closer to a Gaussian and thus simple statistics such as average, STD and SEM should be performed on the log scale. Therefore, for the sake of consistency we believe that the table should be provide as pEC50, while the text can be referring to linear scale values for the ease of reading.

3) Fig. 1e: Why was ITC not performed with the DeltaF508 NBD1 mutant?

We have tried these experiments but they are technically challenging because high protein concentration is required and F508del tends to precipitates under stirring condition. Therefore these experiments are not reliable.

4) Middle of p.7: Is there a pattern regarding the interacting framework residues of these three nanobodies versus the other nanobodies of the study?

As shown in Supplementary Figure 1, the framework residues are well conserved among the different nanobody families and no obvious pattern can explain their binding properties.

5) Fig. S4: I suggest including the data with the normal NBD1 variant side-by-side with 2PT-NBD1-RE to appreciate the differences.

As suggested, data with 2PT-NBD1 have been added side-by-side with 2PT-NBD1-RE in Supplementary Figure 4b.

6) Fig. 5a lacks positive control (i.e. conditions in which 100 % activity were determined). Was it buffer or an unrelated binder?

We have now added a negative control nanobody in the experiment (thus providing a positive control for ATPase activity).

7) Superimpositions of nanobodies with FL-CFTR. Have the authors also superimposed the nanobodies with outward-facing FL-CFTR having closed NBDs? I specifically wonder about G3a, which is not inhibitory. Does it really not sterically clash with closed NBDs?

This is presented in Supplementary Figure 6a, which indeed shows that it does not clash with the NBD1.

Summary:

The manuscript by Sigoillot et al. is well written and provides new insights into how NBD1 may be stabilized through nanobody binding. Atomic detail of the binding interfaces between the stabilizing nanobodies and isolated NBD1 provide valuable information on target protein surfaces in NBD1 for the development of further stabilizing small molecule therapeutics. The manuscript ends by proposing a novel conformational dynamic of NBD1 in the full-length protein, in which NBD1 'undocks' from TMD1.

Comments:

- The final proposed model in which NBD1 'undocks' from TMD1 (Fig. 7) is speculative, as it is inferred mostly from data involving isolated NBD1, crystalized with nanobodies in the absence of the full-length protein. The reader is left wondering if the nanobodies T4 and T8 could bind a different location on the full-length protein that would not require 'undocking'. Could the conformational dynamics that enable

This point was raised by several reviewers, here is our answer:

We are not aware of a single case where a given monoclonal antibody/nanobody would recognize different epitopes (or bind differently) two forms of the same proteins (i.e. domain vs. full-length). As we observe high affinity binding to both isolated NBD1 and full-length CFTR, it seems improbable that the nanobodies would bind differently the two forms. Our high-resolution structures illustrate how the given sequences of each nanobody achieve detailed interactions with specific sites of NBD1 and it expected that these interactions will be achieved in both cases. We observed such behaviour for both T4 and T8, with high affinity binding on both NBD1 and full-length CFTR although the two nanobodies bear different sequences. It thus seems that the unexpected/unlucky scenario of two epitopes is quite unlikely.

However, in order to answer the reviewers' request, we have performed a new experiment to demonstrate that the nanobodies behave identically on isolated NBD1 and full-length CFTR, by taking advantage of the fact that the epitope of T8 requires F508 and that T8 is unable to bind 2PT-F508del-NBD1. In order to solely look at epitope recognition and not protein unfolding, we used a F508del mutant full-length protein stabilized by the 2PT mutations (2PT-F508del-CFTR). This construct is known to mature and properly channel ions (Aleksandrov JMB 2012). Flow cytometry experiments were performed with representative nanobodies (T2a, T8 and G3a). As shown below, T2a and G3a are able to recognize the protein, demonstrating that the overall fold of NBD1 is indeed native in this construct. However, T8 is unable to recognize 2PT-F508del-CFTR, thus demonstrating that F508 is indeed key part of the T8 epitope in the context of full-length CFTR. In our opinion, this demonstrates that the epitopes are indeed similar in isolated NBD1 and in the complete channel.

Left panel: Flow cytometry analysis of nanobodies T2a, T8 and G3a on BHK-21 cells expressing 2PT-F508del showing increased labelling for T2a and G3a, but not T8.

Right panel: Quantification of data illustrated in (h). Average of 3 independent experiments (\pm SEM).

The data are now presented in Figure 4.

nanobody binding occur on immature CFTR during trafficking, rather than on mature CFTR? Further, the

The pull-down experiment (Fig. 4g) shows that T8 binds mostly to fully matured CFTR (band C).

biological relevance of such conformational dynamics is unclear, particularly since T4 and T8 nanobody stabilization of CFTR is measured for a detergent solubilized state and in semi-permeabilized cells. More conclusive evidence for such large-scale motions in full-length CFTR should be provided (e.g. cross-linking, mutagenesis, or structural studies).

We are very much keen to provide direct structural evidences to support our hypothesis, for example using single-molecule FRET experiments on double-cysteine mutants in a Cys-less background. These are very challenging experiments that are ongoing but will require careful optimization. We believe that this goes beyond the scope of this current communication paper and will likely deserve a dedicated manuscript further down the road.

- The main findings in the paper appear to be the identification of specific regions of NBD1 that could contribute to stabilization of the full-length protein if bound by nanobodies, peptide mimetics, or small molecules. As such, the title of the paper, citing "conformational dynamics", is perhaps misleading.

As discussed below, we are convinced that the data support unexpected dynamics of CFTR. While the identification of stabilizing nanobodies opens a new strategy to design CF therapeutics, we believe that both of these important points deserve to appear in the title.

- Additionally, there are areas in the text where data for some nanobodies is unexplainably missing. For example, Fig. 4 panel f, immunoblots of full-length CFTR pull-downs are not provided for nanobodies D12, T27, and T4. Fig. 5, panel a and b, ATPase activity is missing for full-length CFTR and T4 nanobody. Although the nanobodies T4 and T8 are appreciably similar, data should be provided for all nanobodies.

In all our assays, nanobodies D12, T2a,T27 (on the one hand) and nanobodies T4, T8 (on the other hand) behave similarly and bind the same epitope. Therefore, we have selected T2a, T8 and G3a as representative of each of the three "families", especially for experiments where the material was scarce. In order to clarify this for the reader, Figure 4 now only shows data for these 3 representative nanobodies (for coherence, Fig 6 and Supp6 also show the structures of T2a, T8 and G3a). In order to re-assure the reviewer, we have now added data of the missing nanobodies for the pull-down experiment. The FACS and pulldown experiments for D12, T27 and T4 are now presented as supplementary Figure. We now also have added the missing nanobodies to Figure 5 (ATPase assay and thermoprotection). As expected D12 and T27 behave like T2a and T4 behaves like T8.

- While the main findings of the manuscript are interesting and will benefit the CFTR field, the conclusions in the present version of the manuscript seem ultimately too speculative for publication in Nature Communications.

We respectfully disagree with the reviewer. We have been analysing these data for many years now, discussed it with colleagues and collaborators, and presented them at various meetings. Together with the new experiments requested for this reviews process and the re-analysis of older published data, the motion of NBD1 clearly seems like the most likely explanation for our observations. We agree that it is unconventional and goes against the preconceived idea of rigid body motion, but data from the CF field and other fields indicate that conformational dynamics is far more complex than previously expected. As such we feel entitled to present the uncoupling hypothesis in the Discussion section of our manuscript and in its title.

In this manuscript, the authors describe work leading to the identification and characterization of a series of nanobodies from llamas, developed by phage display screening and selection. Antigen used for immunization, importantly, was a version of wildtype nucleotide-binding domain 1 (NBD1) of CFTR that included a series of mutations shown by this group and others to introduce thermal stability. In characterizing the six selected nanobodies, the authors used a variety of biophysical techniques and compared the binding to the same wildtype NBD1 protein and to NBD1 bearing the most common CF-causing mutation, F508del; importantly, all of these proteins still included the stabilizing mutations, meaning that binding data are not necessarily reflective of the truly wildtype target. The authors then also showed that several of the nanobodies bind to full-length CFTR. First, they used purified full-length CFTR in ELISA assays, and then they used full-length CFTR in situ in BHK cells permeabilized to enable access of the nanobodies to their intracellular target sites.

However, the reader is not told whether the versions of full-length CFTR used in these latter experiments also includes the stabilizing mutations

All of the experiments on FL protein involved the wt human sequence with the exception of Figure 5c-e. While it was briefly mentioned in the text, we apologize for not properly labelling the experiment involving this construct. This is now corrected (wt-CFTR vs. stab-CFTR).

The authors have accomplished some very important and very interesting research as reported in this manuscript. However, they are quite ambitious in the stated implications of their research. Claiming that a small-molecule derivative of these nanobodies could be a therapeutic without showing any physiological correction of trafficking of mutant CFTR, and while also showing a significant decrease in ATPase activity when the nanobodies interact with the protein, is misleading and contrary to the potential of these tools as therapeutics.

The CF literature shows that thermal stabilization of F508del-CFTR can overcome the deleterious effect of the mutation. Our study show that thermal stabilization can be achieve by binding of an exogenous molecule. The reviewer is correct that the decrease in ATPase activity is suggestive that these nanobodies may not have direct therapeutic potential, and this is stated in the manuscript. Indeed we do not pretend that the nanobodies are putative therapeutic tools themselves but rather suggest (in the Discussion) that they could provide a route for structure-based design of stabilizers (which are still needed). Therefore we believe that our text is not misleading in any way.

Also, claiming that NBD1 likely moves away from the TMDs based on a superimposed version of the NBD1-nanobody crystal structure on the full-length crystal structure is premature.

The undocking of NBD1 is presented, in the Discussion, as the most likely scenario (in our opinion). We believe that this is consistent with both our data and the literature (see below).

Most of the data are technically sound.

In general, the authors do not do a good job of explaining the very surprising results regarding the availability of the apparent binding sites in the full-length protein. There has been quite a lot of discussion in the CFTR community regarding the question of whether the two NBDs fully disengage EVER in the physiological context.

The ATP free cryo-EM structure (Liu Cell 2017) is now regarded as a relevant conformation of CFTR and show disengagement of the two NBDs relative to each other. While the conformation could be transient it is likely to represent a biological state, in exchange with the nucleotide-bound state also solved by cryo-EM (Zhang Cell 2017).

Not only does their proposed model (Figure 7) suggest that the NBDs do fully disengage, by a degree large enough for the D12/T2a/T27 nanobodies to reach their binding sites, but then for the T4/T8

As shown in Figure 6, the cryo-EM structure of CFTR provides enough space for D12/T2a/T27 to bind between the NBDs. Considering that this disengaged structure demonstrates that this state is indeed accessible (albeit maybe transiently) to CFTR, our data further support the idea that the disengaged state is populated in a cellular context.

Major

comments:

1. Figure 1. It is really disturbing how often the authors end their studies at $n=2$. To calculate standard deviation with $n=2$ is a huge stretch.

We performed additional experiments for ELISA, ITC assays in order to accumulate $N \geq 3$ and modified Figure 1e accordingly.

2. The authors never address the question of whether NBD1 peptides used as immunogens, or in the biophysical assays upon which the paper is based, remain monomeric. If the peptides are forming dimers, interpretations would be greatly changed. This also leads one to question why all of the

For both immunization and characterization, we did not use peptides but, as specified in the Methods, the entire NBD1 domain. This domain remained monomeric even at high concentration (several mgs/ml) as measured by size-exclusion chromatography. Llamas were immunized with NBD1 domain at 1 mg/ml (thus initially monomeric) that got rapidly diluted, so we are confident that the monomeric state was preserved.

biophysical assays were done with only NBD1 instead of with NBD1 plus NBD2 peptides, which would be more physiologically relevant.

As the antigen was solely NBD1, all the epitopes are contained within NBD1 and there is no reason to believe that the nanobodies may interact with NBD2. As discussed for D12/T2a/T27, they may interfere with NBD1:NBD2 association, but that was not the purpose of the presented experiments. Furthermore, the use of fused NBD1-NBD2 constructs remain controversial and thus it may not be feasible to reliably measure the effect of the nanobodies on NBD1:NBD2 association using such constructs.

3. Similarly, NBD1 almost certainly has nucleotide bound most of the time, given cellular conditions. Why were biophysical studies done without ATP or a non-hydrolyzable analog?

As stated in the Methods section, 2 mM ATP + 3 mM $MgCl_2$ was used throughout.

4. To achieve a crystal structure for some of their constructs the authors had to treat their samples with proteases. This reviewer is nervous about that, as that is a pretty significant change to the NBD1 structure, even if the changes were all far from the binding site.

3 of the 5 structures presented were obtained in the presence of proteases. As discussed in the manuscript, the protease resistant segments were virtually identical ($RMSD < 1 \text{ \AA}$) to the corresponding segments in non-protease-treated structures as well as in the structures of human NBD1 published by others.

5. The authors need to add a supplemental figure confirming that the negative control nanobody actually does not bind to CFTR.

Figure 4a,b now shows data where the signal of the negative control nanobody has been subtracted.

6. At the end of the section "Interaction of nanobodies with full-length CFTR," the authors note that the nanobodies cause a drastic decrease in ATPase activity. These data would suggest that these nanobodies (or some small-molecule derivative based upon them) would be bad candidates for therapeutics. Hence, CFTR ion channel activity (WT and F508del) should be studied in the presence of these nanobodies. For purified protein, the Riordan lab has strengths in bilayer reconstitution studies. For the cellular context, excised patch recordings can be used to prevent the need for permeabilizing the cell membrane. It is very surprising that there were no functional studies included beyond ATPase assays.

As discussed in the manuscript and as underlined by the reviewer, the inhibitory effect of nanobodies on ATPase activity may indicate that per se they might be inadequate therapeutic candidates. What we propose in the manuscript is to use the structural characterization of the NBD1:nanobody complexes to understand the molecular basis of stabilization and use that knowledge for subsequent development of small-molecules that would be free of the large steric constraints found with nanobodies. Characterizing the effect of the nanobodies on channel activity will be interesting in the perspective of using them as tool compound to better understand how the channel work, or to lock specific states for structural studies. While this will be technically challenging, we are currently planning such experiments in the forthcoming future. However we strongly feel that these would go beyond the scope of the present communication.

7. In this same section, the authors showed in Figure 4a that nanobodies T2a, T8, and G3a bind purified CFTR protein with high affinity. Did they not test the other three nanobodies in similar experiments? Seems odd that these data were not provided.

In all our assays, nanobodies D12, T2a and T27 behave similarly and indeed they bind the same epitope. Similarly, T4 and T8 share the same epitopes and behave identically in our assays. Therefore, we have selected T2a, T8 and G3a as representative of each of the three "families", especially for experiments where the material was scarce. In order to clarify this for the reader, Figure 4 now only shows data for these 3 representative nanobodies. In order to re-assure the reviewer, we have now added data of the missing nanobodies for the pull-down experiment. The FACS and pull-down experiments for D12, T27 and T4 are now presented as supplementary Figure 5. We now also have added the missing nanobodies to Figure 5 (ATPase assay and thermoprotection). As expected T4 behaves like T8 and D12 and T27 behave like T2a.

8. Importantly, the authors need to test the ability of the nanobodies to bind full-length CFTR under more physiological conditions. In the flow cytometry experiments described in Figure 4, the cells have been permeabilized (to enable access of the nanobodies to their cytoplasmic binding sites) and the cytoplasm is flushed with PBS plus detergent. We have no idea how this will affect the conformation of CFTR. Given the very surprising results presented in this paper, as noted above, this experiment needs to be repeated under physiological conditions that control the state of CFTR such as: inclusion of divalent cations, inclusion of ATP, phosphorylation of CFTR.

The reviewer makes an important point regarding modulation of CFTR dynamics by biochemical parameters such as cations, ATP or phosphorylation. Indeed, we are currently investigating how the conformational landscape of CFTR is affected by these. While these are ongoing investigations, we are happy to share with the reviewer preliminary data showing that the presence of ATP and cations does not prevent nanobody binding, including for T8 as measured by flow cytometry.

Flow cytometry experiments performed as described in the manuscript except that, in addition to PBS (PBS : NaCl 137 mM, KCl : 2,7 mM, Na₂HPO₄.H₂O : 10 mM, KH₂PO₄ : 2 mM, pH 7.4) and 6% FBS, cells were permanently incubated in presence of 1 mM ATP and 1 mM MgCl₂.

Interestingly, while Forskolin-induced phosphorylated does not prevent binding by any of the tested nanobody but we do observed some reduction in apparent labelling

Flow cytometry experiments performed as described in the manuscript except that cells were either incubated with vehicle or treated with 10 μM forskolin, 100 μM DiBu-cAMP and 1 mM IBMX in order to induce phosphorylation of CFTR (phosphorylation was verified by western blot).

While these results need further investigations that will go beyond the scope of the present study, we believe that they demonstrate the relevance of the data as presented in the manuscript and support the overall conclusions

9. Page 9, paragraph 3. The authors found that the interaction of nanobodies D12, T2a, and T27 with NBD1 overlaps with the location of the regulatory extension (RE), although the RE was removed from the NBD1 construct used for immunization. When the authors tested the ability of these three nanobodies to bind NBD1 with the RE, the wording used to describe the result is misleading. Here, the authors state that they "observe a high-affinity binding... albeit with a decrease in apparent EC₅₀," in regards to the nanobodies binding 2PT-NBD1-RE. This sentence is slightly misleading, as a decrease in binding EC₅₀ is directly caused by a decrease in binding affinity. Therefore, it would be more accurate to say that they "observe a decreased but still high binding affinity."

The sentence has been modified as follow: *When we tested whether our nanobodies were able to bind a construct containing the RE (2PT-NBD1-RE), we still observed high-affinity binding for nanobodies D12, T2a and T27, albeit with decrease in apparent EC₅₀ compared to 2PT-NBD1 (Supplementary Fig. 4b).*

10. The authors claim that these nanobodies stabilize F508del, and therefore likely increase trafficking, but never treat CFTR-expressing cells and assess the change in cell surface expression of CFTR. This is vital if the authors want to show that these nanobodies increase trafficking and might actually be informative toward therapeutic development. However, the authors would have to somehow permeabilize the cells to allow the nanobody in to reach their binding sites, or use another delivery method. Certainly, this will be difficult, but needs to be explored.

We do not claim at this stage that the nanobodies stabilize F508del as these experiments have been performed on stabilized F508del-NBD1. We entirely agree that this need to be investigated and we are

currently performing such studies. However, this will be a large scale project that goes beyond the scope of the present manuscript.

11. There are odd cases of missing results. For example, why are there no data in Figure 5 for ATPase activity or ΔT_m for the T4 nanobody? Why are there no data for ITC on the D12 nanobody, especially since D12 appears in 3 of your 4 double nanobody crystallization conditions? Why is there no data for nanoDSF on the G3a or D12? Including those two, considering they are on the ends of the thermal shift data presented in the earlier panel (Figure 5b), would support the findings in both methods.

As explained above we have limited some experiments to the representative nanobodies to allow proper repetition in case of limited material supply. A clear case is Figure 5c-e, where we performed nanoDSF for only one member of each family.

Regarding ITC experiments, nanobody D12 turned out to be unstable under ITC conditions (high protein concentration and high stirring) preventing any reliable interpretation of the data. This nanobody was however properly behaved for the other assays presented in the manuscript.

12. Why does the NBD1+T4 have two thermal shift peaks in Figure 2? Is this partial unfolding of the nanobody or NBD1; and why only for that nanobody? The authors need to address this experimental concern.

We had not provided in the original manuscript the unfolding curves of the isolated nanobodies to avoid overloading of data. The thermofluor assay is well suited here because the signal from NBD1 is strong, while the signal from each nanobody range from moderate (for example signal for T4 is visible on Fig. 2a) to very weak (for example D12), and is always at higher temperatures than the transition of NBD1. Thus the temperature shifts presented do indeed represent the signal coming from SYPRO binding to NBD1. This is also confirmed by the DSC data (Supplementary Fig. 3). However, as requested by the reviewer, we now present the individual unfolding curves and a table reporting melting temperature of each nanobody alone and in presence of NBD1 protein in the Supplementary Figure 3b,c.

Please note that the thermal-shift conditions were optimized for NBD1 and not to test nanobody stability. This assay is protein dependent and optimization of dye and/or protein concentrations may be necessary for optimal assay performance. For example, nanobodies T8 and G3a exhibited a very low fluorescence signal in such conditions. In order to accurately determine melting temperature of all the individual nanobodies we have performed additional experiments at higher concentration of SYPRO Orange dye (5x instead of 2.5x excess). As presented in the new table, we were able to accurately determine T_m for T8 and G3a, and did not observe significant change for the other nanobodies (Supplementary Fig. 3b).

13. The authors mention that papain and subtilisin A were used to generate diffracting crystals, but that the nanobody binding interface with NBD1 was unaffected. Please provide a structural comparison figure to support this evidence.

The conservation of the interface is illustrated in Figure 3. We have now quantified the structural variation for the regions involved in the interactions and measured an RMSD $< 1 \text{ \AA}$. The following sentence was added page 7: *However, the regions involved in binding are highly similar in spite of protease treatment. For example we measured an overall RMSD below 1 \AA for the C_α 's of residues involved in the binding interface shared (see below) by D12 (where no protease was used), T2a (treated with papain) and T27 (treated with subtilisin).*

14. On pg 17, in the Conclusions, the authors make a pretty large logical jump saying that NBD1 probably springs out from the TMD and then comes back to allow T4/T8 nanobodies

to bind. However, this is based on the NBD1-alone crystal structure superimposed on the full-length structure. It is completely possible that the nanobody can force its way in between NBD1 and TMD, or capture a

We respectfully disagree with the reviewer and do not think that a nanobody can “force its way in”. Like other antibodies, nanobodies are developed by the immune system to recognize existing conformation (presented epitope), and thus are expected to bind a population of antigen with epitope directly accessible. A non-zero k_{on} implies that the epitope must be accessible, even transiently in order for the nanobody to bind and shift the equilibrium. Induced-fit model for ligand-receptor interactions proposes that ligands may initially recognize an encounter-state that allow subsequent binding steps to the orthosteric site but this implies that the protein has evolved such beneficial function. However such scenario is highly unlikely here as the sequence of CFTR did not evolve to recognize the nanobodies (and the nanobodies were raised against isolated NBD1). After analysing our structural, biochemical, functional and cellular data for several years, we are convinced that the most likely explanation is an unmasking of the epitope through conformational rearrangement.

In addition, during the reviewing process we re-assessed some previously published data (we are grateful to referee #2 for pointing one on them) and have added a summarizing paragraph in the Discussion:

While undocking of NBD1 from ICL4 may appear surprising, it is supported by previous work. Earlier studies have shown that cysteines introduced in NBD1 (at position 508) and ICL4 (at position 1068) which are separated in the cryo-EM structure by about 7 Å (C_{β} - C_{β} distance) can be efficiently bridged using crosslinkers of lengths ranging from 4 Å to 24 Å, indeed supporting the concept of large domain motion (Serohijos, PNAS 2008). Furthermore, crosslinking these two positions with the short M1M leads to inhibition of channel gating, which can be reverted by reducing agent, suggesting that such domain motion may even be required for proper function. In addition, HDX experiments on the bacterial ABC homodimeric transporter BmrA have shown that the ICD2 peptide (corresponding to ICL4 in CFTR) exchanges extensively with the solvent, indicating that is not permanently buried as observed in crystal structure of homologs (Mehmood PNAS 2012) which suggests that NBD undocking may be happening in other members of the ABC family.

conformation rarely experienced due to the high affinity of the nanobody, thereby creating the false

A rare conformation can be physiologically relevant. As requested by referee #2 we have compared the B_{max} of isolated NBD1 and full-length CFTR for the 3 representative nanobodies. As shown now in Figure 4b, measurements on full-length CFTR indicate that at most 30% of the population is able to recognize the T8 nanobody. We also note that our pull-down data demonstrate that undocked state can be adopted by fully mature protein (Fig. 4g). As such we are confident in stating that this conformation exists (even transiently) in the full-length protein.

impression of a new conformation. It is also possible that the nanobody does not bind in exactly the same way when the rest of the protein is there. These hypotheses are especially possible since the

This point was raised by several reviewers, here is our answer:

We are not aware of a single case where a given monoclonal antibody/nanobody would recognize different epitopes (or bind differently) two forms of the same proteins (i.e. domain vs. full-length). As we observe high affinity binding to both isolated NBD1 and full-length CFTR, it seems improbable that the nanobodies would bind differently the two forms. Our high-resolution structures illustrate how the given sequences of each nanobody achieve detailed interactions with specific sites of NBD1 and it expected that these interactions will be achieved in both cases. We observed such behaviour for both T4 and T8, with high affinity binding on both NBD1 and full-length CFTR although the two nanobodies bear different

sequences. It thus seems that the unexpected/unlucky scenario of two epitopes is quite unlikely. However, in order to answer the reviewers' request, we have performed a new experiment to demonstrate that the nanobodies behave identically on isolated NBD1 and full-length CFTR, by taking advantage of the fact that the epitope of T8 requires F508 and that T8 is unable to bind 2PT-F508del-NBD1. In order to solely look at epitope recognition and not protein unfolding, we used a F508del mutant full-length protein stabilized by the 2PT mutations (2PT-F508del-CFTR). This construct is known to mature and properly channel ions (Aleksandrov JMB 2012). Flow cytometry experiments were performed with representative nanobodies (T2a, T8 and G3a). As shown below, T2a and G3a are able to recognize the protein, demonstrating that the overall fold of NBD1 is indeed native in this construct. However, T8 is unable to recognize 2PT-F508del-CFTR, thus demonstrating that F508 is indeed key part of the T8 epitope in the context of full-length CFTR. In our opinion, this demonstrates that the epitopes are indeed similar in isolated NBD1 and in the complete channel.

Left panel: Flow cytometry analysis of nanobodies T2a, T8 and G3a on BHK-21 cells expressing 2PT-F508del showing increased labelling for T2a and G3a, but not T8.

Right panel: Quantification of data illustrated in (h). Average of 3 independent experiments (\pm SEM).

The data are now presented in Figure 4.

change in T_m is less when T4/T8 is added to FL-CFTR as compared to NBD1 alone. The change in

T_m values and ΔT_m values can hardly be compared between isolated NBD1 and FL protein, especially since different techniques are used.

conformation schematically proposed in Figure 7, to accommodate the generation of a binding site for the T4 nanobody, would likely be energetically very expensive. The authors should include a consideration of how much energy it would take to unfold NBD1 to this extent.

We want to stress that at no point do we suggest unfolding of NBD1 but rather domain motion. Indeed, nanobody recognition requires properly folded NBD1 as their specific epitopes are conformational/discontinued (demonstrated by the crystal structure). Rather, the conformation suggested in Figure 7 requires a simple separation of (folded) NBD1 from the TMD/ICL domain. We understand that the referee asks whether this separation would be energetically possible.

While absolute energy calculation of protein-protein interfaces remains very challenging, a comparative approach is possible. The PISA server can be used for a rough estimation of the interface energy. Using the 5UAK cryo-EM structure of human CFTR at 3.87 Å resolution, the server algorithm measures an interface between NBD1 and the TMD/ICL of 1028 Å² with a ΔG of -15.8 kcal/mol. Of note, Nooren and Thornton (JMB 2003) have described that transient interfaces typically fall in the 600-1500 Å² range.

We have then used a cryo-EM structure of an unrelated protein at a comparable resolution, the recent μ -opioid receptor in complex with the G protein at 3.5 Å resolution (6DDE). For this structure, the PISA server measures an interface of 1217 Å² between the receptor and the G $_{\alpha}$ with a ΔG of -14.3 kcal/mol and an interface of 1213 Å² with a ΔG of -15.4 kcal between the G $_{\alpha}$ and the G $_{\beta}$ subunits. These interfaces are well known to dissociate during the activation cycle. In contrast, the G $_{\beta}$ -G $_{\gamma}$ subunits, which do not dissociate, show an interface of 2007 Å² with a ΔG of -35.7 kcal/mol.

These comparative calculations therefore suggest that the NBD1:TMD/ICL interface is not necessarily permanent and thus supports our model of NBD1 detachment required for T4/T8 binding.

15. The authors talk about how large the binding interface of these nanobodies is (600-1000 angstroms), yet claim a small molecule could be developed to replicate nanobody binding. If this is going to be a claim, the authors need to identify a small section of the binding interface that is vital for stability. This, however, would basically require the development of the proposed small molecule. The binding interface for nanobodies is considerably larger than that of small molecules.

Our manuscript shows how nanobodies stabilize NBD1 and suggests that therapeutic molecules could be developed using the structural information presented. Although there are examples of small molecules compounds with large interface surface area of the protein (e.g. tacrolimus buries 720 Å² in calcineurin) we do not claim that nanobody-inspired therapeutics will require to develop small molecules with binding interfaces of 1000 Å² but rather that bridging distant pockets may prove valuable. There are various examples in the literature of bivalent molecules designed to target two separated pockets (an extreme example are the bivalent dopamine receptor ligands developed by the Gmeiner lab, see Kühhorn J. Med. Chem. 2011). However, while we (and maybe others) will investigate such possibilities in the future, this goes beyond the scope of the present communication paper.

Minor comments:

1. Page 2: The authors misstate the data on number of mutations. Certainly, >2000 mutations in the CFTR gene have been identified in the human population. However, it is not known what number of these are actually disease-causing mutations. The authors are referred to the CFTR2 database for clarification: <https://www.cftr2.org/>

We thank the reviewer for spotting this inaccuracy. Based on the 312 CF-causing mutations referred by the CFTR2 database, we have change the sentence to :

Over 300 cystic fibrosis-causing mutations have been described in the CFTR gene (see <https://www.cftr2.org/>)

2. Page 10. In the section "Interaction of nanobodies with full-length CFTR," when the authors reference "mature FL-CFTR," it is important for them to explain what that means in relation to immature FL-CFTR, cellular localization, and apparent size on electrophoresis. Not all readers of Nature Communications will be familiar with "mature" as a description.

We have changed this 2 sentences page 10:

Functional mature CFTR being fully glycosylated, electrophoresis allows to separate it from the intracellular immature CFTR. Mature CFTR with complex N-linked oligosaccharide chains migrates at an apparent molecular weight of 170 kDa (historically called "band C") while immature core-glycosylated CFTR runs at a lower molecular weight (named "band B").

3. It is important for the authors to comment on the lack of immunogenicity of nanobodies, as the reader may become confused as to how a biologic could be used as long-term therapy without mounting a huge immune response.

While there has been quite some effort in "humanizing" nanobodies to decrease immune response (see Vincke JBC 2009) for immunotherapy against various diseases, we do not claim here that NBD1-stabilizing nanobodies should be used directly as therapeutics against CF as sending them inside cells of patients will be very challenging (as explained in the 2nd paragraph of the Discussion).

4. It is not optimal to use thermostabilized mutant CFTR for thermostability assays, but it is understandable. It would be wonderful to see these experiments repeated in truly WT constructs.

For isolated NBD1 we have used a stabilized version for obvious practical reasons (yield, stability). We have nevertheless verified that our nanobodies could bind wt-NBD1 by using a non-cleaved Sumo-hNBD1. While the protein is indeed much less stable than the 2PT variant (T_m of 39 °C vs. 44 °C), it can still be stabilized with the different nanobodies by over 10 °C, except nanobody G3a (see data presented below). While this results validate our approach using the 2PT-stabilized version it was impractical to redo our characterization with the isolated wt-NBD1. However, note that most of our characterization of interactions with the full-length protein was done using a wild-type construct (Fig.4, Fig.5 a,b).

Differential scanning fluorescence (DSF) of purified non-cleaved Sumo-hNBD1. The curves represent the first derivative of fluorescence of non-cleaved Sumo-hNBD1 in absence of nanobody (black curves, $T_m = 38.8$ °C) or in presence of 3x-molar excess of nanobody (colored curves). Melting curves of nanobodies alone are depicted as dashed colored curves.

5. Can the authors confirm that no crystals formed for the nanobody D12 or T4 alone, or no good diffracting crystals, or was this not even tested.

In the course of the project we tried a large number of combinations of nanobodies. We focused on the most promising ones in terms of resolution, leading to the structures presented in the manuscript.

6. Please include the original Δ RE-NBD1 nanobody binding data (EC50) in a table with the RE-NBD1 construct nanobody binding data in Supplementary Figure 4 to confirm no change in binding.

7. Please include the experimental condition of the "CTRL" in figure legend for Figure 4f.

The negative control was a non-specific nanobody (not directed against CFTR), this is now specified in the legend.

8. Please include a bar for the WT-FL-CFTR ATPase activity data (100%)

We now include a negative control nanobody (100% ATPase activity).

9. Maintain the color code from Figure 4a-d in Figure 5.

After trying several versions we have now simplified our color coding for clarity.

10. Please provide more explanation of the purification of the thermostabilized human CFTR construct from BHK-21 cells (at least information on whether the construct is tagged or untagged).

We now mention that the CFTR construct is His-tagged.

11. Typos and errors throughout the manuscript

We thank the reviewer and have corrected the typos. The manuscript has been corrected by a scientist whose first language is English.

Reviewers' Comments:

Reviewer #1:

Remarks to the Author:

The authors have made good-faith responses to most of the questions raised in the original review. There are some remaining issues, primarily concerning the strength of the argument for large and dynamic conformational changes in FL-CFTR.

1) The authors ask for evidence from other cases where the binding pose of a Nb is affected when it binds to a domain in the context of a full-length protein. Of course, in some cases, the binding is simply blocked if the epitope is obscured in the full-length protein. However, a quick search revealed a possible example of adaptation: in the case of a broadly neutralizing HIV-specific antibody that targets a membrane-proximal helical epitope, residues from helix H1 that affect affinity were found to be disordered (and thus presumably not to contribute to affinity) for a DPC-embedded antigen. In other words, the epitope may have contracted/shifted, while retaining high-affinity binding (Scientific Reports 6:38177). There may well be other examples in the literature. Does the involvement of F508 in both interactions (with NBD1 and with FL-CFTR) preclude this possibility, given its steric position within the interfaces characterized to date? If so, the argument should be explicitly described. If not, the text should acknowledge this as a possibility.

2) The model shown in Fig. 7 indicates a very substantial displacement of F508 from the interface (panel C). This is illustrative, but it may convey a misleading sense of the extent of domain unpacking required. Based on the modeling of the Nb:NBD1 complex, what is the minimum rotation of the NBD1 (in any direction) that would avoid steric clash? This should be mentioned, if significantly smaller than the illustrated rotation. (Alternatively, the illustration could be adapted to reflect the smaller required change.)

3) As written, the text seems to imply that the ability of cross-linkers with a range of lengths to connect engineered cysteines across the interface means per se that the cysteines are at variable distances. The cross-linkers used in the cited report are flexible and most likely can interact at shorter distances. The loss of functionality when a short cross-linker is used suggests that a conformational rearrangement occurs across the interface, but not that it must create a dramatically longer separation. This can be addressed by straightforward text revisions.

4) Reading the response to the comments of Reviewer 2 about conformational dynamics in the title raised concerns. Even if the postulated Nb-accessible conformation exists, it is a single new conformation. Of course, a new conformation implies that there is a change. However, CFTR is already known to adopt multiple conformations, and the article reveals little information about the kinetics of conformational interconversion. The title and text should be rephrased for greater accuracy. Instead of "Conformational dynamics of CFTR...", the title would more appropriately read "An unexpected conformation of CFTR revealed by stabilizing nanobodies," or something along those lines. Corresponding changes should occur in the text.

Reviewer #2:

Remarks to the Author:

Overall, the authors have addressed the reviewers' concerns in a satisfactory manner.

I agree with the authors that different binding epitopes of the nanobodies for NBD1 and FL-CFTR can be excluded. A specific protein-protein interaction is extremely complex and rare. Hence, there is only one binding epitope on a protein (or several identical epitopes if the protein is a homo-multimer, which is not the case for CFTR).

Further I agree with the authors that the model they draw in Figure 7 is the logical interpretation of their data. I do not see any other possible interpretation. Of course, the interpretation looks a bit adventurous, but this is not a reason to call it wrong or the result of over-interpretation.

I am still not fully satisfied regarding two specific points:

Supplementary Figure 3: Legend of Fig. S3c lacks the color code. I assume that the grey curves correspond to nanobody unfolding alone and the green curve is NBD1 + Nbs. Further, it is not clear whether the Sypro 2.5x or 5x data are shown in (c). Last but not least, I strongly suggest to include also some raw data of the unfolding curves (NBD1 alone, nanobody alone and combo; concentrations not adjusted to get higher signal for nanobody) in order to show that the nanobodies do not strongly contribute to the Sypro signal (as the authors claim in the point-by-point answer). What is shown in (c) is the first derivative of the raw data. From this data I cannot see the absolute contribution of Sypro signal generated with the nanobodies.

Figure 4: The authors now determined Bmax values for nanobody binding. I find it surprising that there is no difference regarding Bmax values of the nanobodies when the nanobodies were immobilized and FL-CFTR titrated (Fig. 4a), whereas the authors observe Bmax differences if FL-CFTR is immobilized and the nanobodies are titrated (Fig. 4b). A further surprise: the KD values seem to be quite different between Fig. 4a and Fig. 4b. The dataset unfortunately lacks negative controls. How can the authors exclude that FL-CFTR sticks to the ELISA plate in an unspecific manner independent from an immobilized nanobody, or in the presence of an unrelated nanobody? The same is true for Fig. 4b: the titration curves of the nanobodies should have been conducted also on empty wells (or even better on wells containing a membrane protein other than FL-CFTR immobilized) to exclude unspecific binding. The curve of an unrelated nanobody would have been desirable (but not mandatory).

Once the authors have addressed these remaining points, I recommend publication of this interesting study in Nature Communications.

Reviewer #3:

Remarks to the Author:

While the manuscript is improved, the authors have still not provided strong evidence for the proposed 'large scale domain motions' for the undocking of NBD1 from ICL4, which they have made the focus of the paper. The rebuttal letter mentions ongoing studies that would address the concerns raised by all four reviewers about the biological relevance and functional impacts of NBD1 undocking; however, the authors have no intention of providing that evidence in the current manuscript. The model proposed in Fig. 7, while potentially very interesting, remains highly speculative.

Without more concrete proof of NBD1 undocking [e.g. structural confirmation, cross-linking studies, functional studies of CFTR in lipid bilayers (absence of detergent)], the manuscript is not suitable for publication in Nature Communications.

Reviewer #1 (Remarks to the Author):

1) The authors ask for evidence from other cases where the binding pose of a Nb is affected when it binds to a domain in the context of a full-length protein. Of course, in some cases, the binding is simply blocked if the epitope is obscured in the full-length protein. However, a quick search revealed a possible example of adaptation: in the case of a broadly neutralizing HIV-specific antibody that targets a membrane-proximal helical epitope, residues from helix H1 that affect affinity were found to be disordered (and thus presumably not to contribute to affinity) for a DPC-embedded antigen. In other words, the epitope may have contracted/shifted, while retaining high-affinity binding (Scientific Reports 6:38177). There may well be other examples in the literature. Does the involvement of F508 in both interactions (with NBD1 and with FL-CFTR) preclude this possibility, given its steric position within the interfaces characterized to date? If so, the argument should be explicitly described. If not, the text should acknowledge this as a possibility.

The reviewer points out an interesting, albeit peculiar, case of structural alteration between two forms of the antigen. However we believe that this specific example also supports our claim of a unique epitope, which is also asserted by referee #2 in her/his latest comment (*"A specific protein-protein interaction is extremely complex and rare. Hence, there is only one binding epitope on a protein"*). Indeed structural analysis of the two structures discussed in the mentioned study (Rujas et al 2016) demonstrates that the epitope itself is overall well conserved (as shown below). The details of the protein-protein interactions are also conserved with side chains even adopting identical rotamers in both sides of the interface (with the exception of W680). The main difference between the two presentations is the destructure of a short N-terminal helix that, as stated by the authors, does not seem to contribute to the interaction. To our knowledge there is no clear example of epitope change between different antigen presentations.

Considering the complexity of the NBD1:T8 interaction we cannot envisage a different mode of binding than the one observed in the crystal structure of the isolated complex and this is supported by the lack of binding of the FL-CFTR-F508del mutant.

2) The model shown in Fig. 7 indicates a very substantial displacement of F508 from the interface (panel C). This is illustrative, but it may convey a misleading sense of the extent of domain unpacking required. Based on the modeling of the Nb:NBD1 complex, what is the minimum rotation of the NBD1 (in any direction) that would avoid steric clash? This should be mentioned, if significantly smaller than the illustrated rotation. (Alternatively, the illustration could be adapted to reflect the smaller required change.)

While there is so far no clear evidence as where or how should NBD1 rotate, the motion must be large enough to unmask F508 and allow the binding of nanobody T4 or T8. The purpose of the cartoon presented in Figure 7 was to illustrate such motion although indeed the exact extent is not yet known. In order to mitigate a possible misperception of an exaggerated displacement, we have now remade the cartoon with a more limited motion of NBD1 (for consistency now with T8 nanobody). Please note that the cartoon is based on the actual EM and crystal structures of CFTR and NBD1:Nb complexes, and thus that the relative scales are meaningful, a much smaller opening would not provide enough space to dock the nanobody.

3) As written, the text seems to imply that the ability of cross-linkers with a range of lengths to connect engineered cysteines across the interface means per se that the cysteines are at variable distances. The cross-linkers used in the cited report are flexible and most likely can interact at shorter distances. The loss of functionality when a short cross-linker is used suggests that a conformational rearrangement occurs across the interface, but not that it must create a dramatically longer separation. This can be addressed by straightforward text revisions.

We have now reworded this paragraph to soften the interpretation of the crosslinking data in the light of the proposed model.

4) Reading the response to the comments of Reviewer 2 about conformational dynamics in the title raised concerns. Even if the postulated Nb-accessible conformation exists, it is a single new conformation. Of course, a new conformation implies that there is a change. However, CFTR is already known to adopt multiple

conformations, and the article reveals little information about the kinetics of conformational interconversion. The title and text should be rephrased for greater accuracy. Instead of "Conformational dynamics of CFTR...", the title would more appropriately read "An unexpected conformation of CFTR revealed by stabilizing nanobodies," or something along those lines. Corresponding changes should occur in the text.

The term "dynamics" typically implies motion, which is required here to enable binding of nanobody T4 or T8. We do not claim to have determined the precise parameters of the underlying motion, such as kinetics. On the other hand, our study does not establish that the T4/T8-bound state corresponds to a single conformation (in fact an undocked state might be more dynamic and flexible than a docked state). Thus, in our opinion the term "Dynamics" remains more general and thus more appropriate for the title, but we would be open to discuss this specific semantic point with the editor.

Reviewer #2 (Remarks to the Author):

Supplementary Figure 3: Legend of Fig. S3c lacks the color code. I assume that the grey curves correspond to nanobody unfolding alone and the green curve is NBD1 + Nbs. Further, it is not clear whether the Sypro 2.5x or 5x data are shown in (c). Last but not least, I strongly suggest to include also some raw data of the unfolding curves (NBD1 alone, nanobody alone and combo; concentrations not adjusted to get higher signal for nanobody) in order to show that the nanobodies do not strongly contribute to the Sypro signal (as the authors claim in the point-by-point answer). What is shown in (c) is the first derivative of the raw data. From this data I cannot see the absolute contribution of Sypro signal generated with the nanobodies.

We thank the reviewer for spotting the color coding error and have corrected it. We also now provide the data as raw fluorescence signal (see below). Please note that for T27 the signal of NBD1-T27 complex is not missing but overlapping with T27 alone.

Figure 4: The authors now determined B_{max} values for nanobody binding. I find it surprising that there is no difference regarding B_{max} values of the nanobodies when the nanobodies were immobilized and FL-CFTR titrated (Fig. 4a), whereas the authors observe B_{max} differences if FL-CFTR is immobilized and the nanobodies are titrated (Fig. 4b).

In the case of immobilized CFTR only a subset of the antigen are competent to bind T8, thus the apparent B_{max} correspond to this subpopulation of CFTR (hypothesized to have an undocked NBD1). When T8 is immobilized, all the sites can be eventually occupied once enough CFTR is provided. We thus do not see this as surprising result.

A further surprise: the KD values seem to be quite different between Fig. 4a and Fig. 4b. The dataset

We agree with the reviewer that the difference in Kd is unexpected and we cannot provide a simple model to explain it at this stage. However, this is the data coming from three independent experiments and thus reliable.

unfortunately lacks negative controls. How can the authors exclude that FL-CFTR sticks to the ELISA plate in an unspecific manner independent from an immobilized nanobody, or in the presence of an unrelated nanobody? The same is true for Fig. 4b: the titration curves of the nanobodies should have been conducted also on empty wells (or even better on wells containing a membrane protein other than FL-CFTR immobilized) to exclude unspecific binding. The curve of an unrelated nanobody would have been desirable (but not mandatory).

There seems to be some misunderstanding. We are very much aware of the possible caveats of ELISA measurements and each experiment was performed with appropriate negative controls (i.e. a non-relevant nanobody). As stated in the legend of figure 4 "For both (a) and (b) Data were normalized to maximal response of T2a after subtraction of the signal from the negative control nanobody".

Reviewers' Comments:

Reviewer #1:

Remarks to the Author:

The revised figure is nicely drawn. There are three remaining points. Rather than addressing the issues broadly, I attempted in the following paragraphs to focus specifically on their relevance to the claims that the authors make in this manuscript.

1) The first concerns the epitope. The authors claim that the epitope observed in NBD1:Nb co-crystals is buried in the full-length structure. However, they do not describe how completely it is buried. The point about the HIV-specific antibody is that binding was preserved even though part of the epitope had been obscured. Key question: is *part* of the NBD1 epitope exposed in any of the known full-length structures of CFTR, and if so, how can the authors exclude the possibility that the nanobody interacts with that residual epitope? If the epitope is 90% buried in all known conformations, I am comfortable with the authors' claims on this point. If it's 40% buried in one conformation, then perhaps the remaining 60% accounts for the retention of binding. How can they exclude that possibility?

2) The second concerns the cross-linkers. It is claimed that the published data (ref. 34) "could agree with large domain motion." The published data could also agree with a small domain motion (the M1M data mean only that the interface needs to be a few angstroms shorter than the observed distance of 7 Å). If the authors strike the word "large," this would be fine. This point is important, because the size and openness of the necessary interface arrangement is at issue in the next point.

3) The third concerns the use of the word dynamics. The authors claim in the title and elsewhere that the stabilizing nanobodies reveal conformational dynamics.

What they really have found is evidence for one conformation not previously anticipated. The second-to-last sentence in the abstract gets it right: "...our data uncover *a novel conformation** of CFTR..." (note the singular noun). This conformation involves the opening of the NBD1:ICL4 interface, suggesting that the interface is more flexible than previously expected. So far, so good.

However, they sometimes make a far bolder claim, for example, in the last sentence of the introduction: "... the location of several epitopes demonstrates that CFTR must be able to adopt *conformations** ...further establishing that CFTR is a highly dynamic protein, even under a normal physiological regime." (plural noun)

Since CFTR has long been known to have many functional states (phosphorylated or not; different nucleotides bound...), the addition of a single state to the known ensemble does not provide evidence for a new kind of conformational dynamics. It just slightly broadens the universe of conformations. Implicitly, the authors assume that the new conformation must be less compact and more mobile. That's certainly how they drew their first version of Figure 7 and it's embedded in their citation of cross-linking data in support of a "large" conformation change (as discussed above in point 2). However, they present no evidence for the dynamic or open nature of this new conformation.

While dynamics are detected with respect to the NBD1:ICL4 interface, it is too speculative to claim a significant increase in protein dynamics overall. In the spirit of trying to get to an end point, here's a suggestion:

Title: "Domain-interface dynamics of CFTR..."

Last sentence of the abstract: "this unexpected interface rearrangement..."

Last sentence of the introduction: "...the locations of several epitopes demonstrate that CFTR must be able to adopt at least one additional conformation...consistent with the view that CFTR is a highly dynamic protein, even under a normal physiological regime."

Reviewer #2:

Remarks to the Author:

The authors have now addressed the last remaining questions and the manuscript has in my view reached high technical standards as well as it provides truly novel molecular insights. I therefore recommend publication of this manuscript in Nature communications.

Reviewer #1 (Remarks to the Author):

1) The first concerns the epitope. The authors claim that the epitope observed in NBD1:Nb co-crystals is buried in the full-length structure. However, they do not describe how completely it is buried. The point about the HIV-specific antibody is that binding was preserved even though part of the epitope had been obscured. Key question: is *part* of the NBD1 epitope exposed in any of the known full-length structures of CFTR, and if so, how can the authors exclude the possibility that the nanobody interacts with that residual epitope? If the epitope is 90% buried in all known conformations, I am comfortable with the authors' claims on this point. If it's 40% buried in one conformation, then perhaps the remaining 60% accounts for the retention of binding. How can they exclude that possibility?

The reviewer suggests that if a part of the epitope remains accessible then the nanobody could still achieve binding (albeit maybe without a lower efficiency). The usual way to measure accessibility of a protein domain/epitope is by measuring solvent accessibility area (ASA), eg. the ability of water molecules to access the part(s) of the protein under scrutiny. We want to stress that binding to the epitope requires more than accessibility of solvent because enough room must be provided to the entire 15 kDa immunoglobulin domain of the nanobody. Nevertheless, we have performed the required calculation and they indeed show very important burying of the interfaces.

In summary, we are confident that the epitope is not accessible to the T4 or T8 nanobodies considering that:

-The ASA calculation (computed with VADAR) shows that accessibility of the epitope (defined as residues within 3.5 Å of the nanobody) is reduced by 61% and 73% buried in the non-phosphorylated (PDB 5UAK) and phosphorylated (6MSM) forms of full-length human CFTR compared to the isolated NBD1 domain.

-F508 becomes 97% (phosphorylated) and 80% (non-phosphorylated) buried when comparing isolated NBD1 and full-length CFTR, indicating that it will not be accessible to the nanobodies in the conformation observed in the cryo-EM structures. Please bear in mind that F508 is absolutely required for binding of T8 and T4. To underline this important point, we have added the following sentence in the Discussion:

Moreover, while F508 is completely solvent exposed in the isolated NBD1 domain, it becomes buried in the NBD1-ICL interface observed in the cryo-EM structures (and thus not available for the nanobodies), while our data have demonstrated that interaction with F508 is strictly required for binding by T4 or T8.

-More importantly, as stated on page 17 of the manuscript: "T4 and T8 completely overlap with the position of the coupling helix of the ICL4, and also with ICL1 and surrounding helices." So even if the epitope would be partly accessible (e.g. to the solvent) the rest of the protein completely prevents this binding mode of the nanobody. This is illustrated in Figure 6 and Supplementary Figure 6. To further our point, we provide two more snapshots below.

The human CFTR cryo-EM structure (PDB 5UAK) is shown in blue (N-term) and green (C-term) ribbons (left and right panels are rotated 90°). The T8 nanobody structure (purple) was placed by superimposing the coordinates of NBD1 from the NBD1-T8 complex and from the structure of human CFTR (RMSD <math><1.1 \text{ \AA}</math>). The extensive steric overlap between the nanobody and the helical loops from CFTR is clearly visible. Such a binding mode is thus not sterically possible.

2) The second concerns the cross-linkers. It is claimed that the published data (ref. 34) "could agree with large domain motion." The published data could also agree with a small domain motion (the M1M data mean only that the interface needs to be a few angstroms shorter than the observed distance of 7 Å). If the authors strike the word "large," this would be fine. This point is important, because the size and openness of the necessary interface arrangement is at issue in the next point.

We have removed the word "large" in the sentence.

3) The third concerns the use of the word dynamics. The authors claim in the title and elsewhere that the stabilizing nanobodies reveal conformational dynamics.

What they really have found is evidence for one conformation not previously anticipated. The second-to-last sentence in the abstract gets it right: "...our data uncover *a novel conformation* of CFTR..." (note the singular noun). This conformation involves the opening of the NBD1:ICL4 interface, suggesting that the interface is more flexible than previously expected. So far, so good.

However, they sometimes make a far bolder claim, for example, in the last sentence of the introduction: "... the location of several epitopes demonstrates that CFTR must be able to adopt *conformations*...further establishing that CFTR is a highly dynamic protein, even under a normal physiological regime." (plural noun)

Since CFTR has long been known to have many functional states (phosphorylated or not; different nucleotides bound...), the addition of a single state to the known ensemble does not provide evidence for a new kind of conformational dynamics. It just slightly broadens the universe of conformations. Implicitly, the authors assume that the new conformation must be less compact and more mobile. That's certainly how they drew their first version of Figure 7 and it's embedded in their citation of cross-linking data in support of a "large" conformation change (as discussed above in point 2). However, they present no evidence for the dynamic or open nature of this new conformation.

While dynamics are detected with respect to the NBD1:ICL4 interface, it is too speculative to claim a significant increase in protein dynamics overall. In the spirit of trying to get to an end point, here's a suggestion:

Title: "Domain-interface dynamics of CFTR..."

Last sentence of the abstract: "this unexpected interface rearrangement...",

Last sentence of the introduction: "...the locations of several epitopes demonstrate that CFTR must be able to adopt at least one additional conformation...consistent with the view that CFTR is a highly dynamic protein, even under a normal physiological regime."

We agree with the proposed changes and have modified the manuscript accordingly. The title is thus now : ***Domain-interface dynamics of CFTR revealed by stabilizing nanobodies.***

Reviewers' Comments:

Reviewer #1:

Remarks to the Author:

The latest modifications address all remaining concerns.